# Injury-induced intestinal stem cell renewal requires capillary morphogenesis gene 2

Lucie Bracq [ID][1][✉], Audrey Chuat [ID][1], Béatrice Kunz[1], Olivier Burri[2], Romain Guiet [ID][2], Julien Duc[3], Nathalie Brandenberg[1] & F Gisou van der Goot [ID][1][✉]

## Abstract

**Patients with the rare genetic disorder Hyaline Fibromatosis Syndrome (HFS) often succumb before 18 months of age due to severe diarrhea. As HFS is caused by loss-of-function mutations in the gene encoding capillary morphogenesis gene 2 (CMG2), these symptoms highlight a critical yet unexplored role for CMG2 in the gut. Here, we demonstrate that CMG2 knockout mice exhibit normal colon morphology and no signs of inflammation until the chemical induction of colitis. In these conditions, the colons of knockout mice do not regenerate despite previously experiencing similarly severe colitis, due to an inability to replenish their intestinal stem cell pool. Specifically, CMG2 knockout impairs the transition from fetal-like to Lgr5+ adult stem cells, which is associated with a defect in ß-catenin nuclear translocation. Based on our findings, we propose that CMG2 functions as a context-specific modulator of Wnt signaling, essential for replenishing the pool of intestinal stem cells following injury. This study provides new insights into the molecular mechanisms underlying lethal diarrhea in HFS and offers a broader understanding of fetal-like regenerative responses.**

**Keywords** Lgr5; Regeneration; Fetal-like Stem Cells; Colitis; Wnt Signaling
**Subject Categories** Digestive System; Genetics, Gene Therapy & Genetic Disease

See also: WI Lencer

## Introduction

Hyaline Fibromatosis Syndrome (HFS; OMIM #228600) is a rare autosomal disorder that, in its most severe form, leads to death before 18 months of age due to intractable diarrhea or recurrent pulmonary infections (Casas-Alba et al, 2018). Despite the critical nature of these symptoms, there are currently no effective interventions, primarily because the underlying molecular mechanisms remain poorly understood. The defining feature of HFS is however the abnormal accumulation of hyaline material, particularly collagen VI (Bürgi et al, 2017), in the skin, which results in large subcutaneous nodules, gingival hypertrophy, and painful joint contractures (Deuquet et al, 2011; Denadai et al, 2012; Urbina et al, 2004; Slimani et al, 2011; Cozma et al, 2019; Hanks et al, 2003; Dowling et al, 2003). HFS is caused by mutations in the Capillary Morphogenesis Gene 2 (CMG2), with evidence showing that CMG2 regulates extracellular collagen VI levels. However, the severe diarrheal symptoms and associated intestinal lymphangiectasia, often culminating in fatal protein-losing enteropathy (Casas-Alba et al, 2018), suggest that CMG2 may play a distinct and crucial role in intestinal physiology that has not yet been characterized. This study aims to address this critical gap by elucidating the role of CMG2 in the gut and uncovering the mechanisms responsible for the devastating intestinal symptoms observed in severe HFS patients.

CMG2, also known as ANTXR2, is a type I membrane protein featuring an extracellular von Willebrand A (vWA) domain, that can bind extracellular matrix components, including laminin and types IV and VI collagen (Bell et al, 2001; Bürgi et al, 2017), as well as the actin cytoskeleton and actin modulators intracellularly (Bürgi et al, 2020). One of the documented roles of CMG2 is to control the turnover of collagen VI by mediating its intracellular uptake and degradation in lysosomes (Bürgi et al, 2017). As a consequence, upon CMG2 loss of function, multi-systemic deposition of collagens, in particular Collagen VI, occurs in HFS patients (Bürgi et al, 2017; Reeves et al, 2012; Bürgi et al, 2020; van Rijn et al, 2020). CMG2 may, however, have roles other than controlling the abundance of the ECM. In particular, while the intestine of two infants with severe HFS showed accumulation of ECM and in particular collagen VI, no overt primary defect in the function of the intestinal epithelium was observed based on in vitro culture of patient cells either as intestinal epithelial layers or gut organoids, although the presence of abnormal intercellular blisters could be observed (van Rijn et al, 2020).

We therefore speculate that CMG2 plays a role in crucial processes such as homeostatic cellular turnover and intestinal regeneration. The YAP and Wnt signaling pathways are central to these processes (Meyer et al, 2022; Deng et al, 2018; Guillermin et al, 2021), and several studies have suggested links between CMG2 and these pathways, most notably the Wnt pathway (Bürgi et al, 2017; Castanon et al, 2020; Abrami et al, 2008; Liu et al, 2023;

[1]Global Health Institute, School of Life Sciences, EPFL, Lausanne, Switzerland. [2]BioImaging and Optics Core Facility, School of Life Science, EPFL, Lausanne, Switzerland. [3]Nexco Analytics, EPFL Innovation Park, Lausanne, Switzerland. [✉]E-mail: lucie.bracq@epfl.ch; gisou.vandergoot@epfl.ch

Ji et al, 2018; Castanon et al, 2013). The normal rapid cellular turnover of the intestinal epithelium is controlled by the interplay between Wnt, Notch, and YAP/Taz signaling pathways (Hageman et al, 2020). After intestinal injury, intestinal stem cells (ISCs) are depleted (Tetteh, 2016; Tian, 2011; Yui, 2018; Murata et al, 2020; Ayyaz et al, 2019; Harnack et al, 2019; Metcalfe et al, 2014), triggering a regenerative response to restore tissue integrity (Hageman et al, 2020; Barker, 2014; Meyer et al, 2022; Rees et al, 2020; de Sousa E Melo and de Sauvage, 2019). Recent research indicates that this regeneration involves dedifferentiation and a fetal-like regenerative response (Tetteh, 2016; Yui, 2018; Murata et al, 2020; Meyer et al, 2022; Rees et al, 2020; de Sousa E Melo and de Sauvage, 2019; Nusse et al, 2018; Higa et al, 2022; van Es et al, 2012; Jadhav et al, 2017; Castillo-Azofeifa et al, 2019; Ishibashi et al, 2018; Yan, 2017; Buczacki et al, 2013; Yu et al, 2018; Jones et al, 2019), a process highly dependent on YAP and Wnt signaling interactions (Yui, 2018; Sprangers et al, 2021; Gregorieff et al, 2015). However, the precise mechanisms governing this regenerative response remain incompletely understood.

In this study, we utilized a knockout mouse model to investigate the role of CMG2 in the colon under both normal and regenerative conditions. While no abnormalities were observed in the intestinal architecture or function under homeostatic conditions, the scenario changed following chemically induced colitis (Yang and Merlin, 2024; Chassaing et al, 2014). *Cmg2* knockout (*Cmg2*^KO) mice failed to restore intestinal integrity, underscoring a critical role for CMG2 in regeneration. Our analysis showed that CMG2 is not involved in the initial fetal-like reversion phase, but is essential to replenish the Lgr5+ intestinal stem cell (ISC) pool—an important Wnt-dependent step for epithelial proliferation and barrier restoration. These findings highlight CMG2's vital role in Wnt-driven ISC replenishment during intestinal regeneration, providing new insights into the transition from fetal-like to adult ISC states, as well as molecular understanding of the chronic diarrhea seen in severe HFS patients.

# Results

Despite intestinal alterations being the most severe symptoms observed in full CMG2 loss-of-function HFS patients, the expression pattern of CMG2 in the gut has not been specifically investigated. To address this gap, we first analyzed publicly available spatial transcriptomics and scRNA-seq datasets to identify which cell types express Cmg2 across different gut regions (Mayassi et al, 2024b). Spatial transcriptomic data from the mouse small intestine and colon revealed that Cmg2 is broadly expressed throughout the gut, including in the muscular, crypt, and epithelial layers (Fig. 1A–C). To validate these findings, we performed RNAscope in situ hybridization targeting Cmg2 in the duodenum and colon of wild-type mice. The observed expression pattern was consistent with the spatial transcriptomics data (Fig. 1D). We next analyzed a single-cell RNA-seq dataset to assess cell-type-specific expression in the mouse colon (Fig. 1E–G). Cmg2 was detected at varying levels across multiple cell types, including epithelial cells such as enterocytes and intestinal stem cells, and mesenchymal cells, notably fibroblasts.

Interestingly, Tem8, which encodes a protein highly homologous to CMG2 that shares it's function of anthrax toxin receptor, displayed a much more restricted expression pattern, being confined primarily to fibroblasts and mural cells, and notably absent from epithelial cells (Appendix Fig. S1A,B). This differential expression highlights a potentially unique and epithelial-specific role for CMG2 in maintaining intestinal homeostasis. Consistently, no intestinal symptoms have been reported for patients suffering from GAPO syndrome (OMIM 230740), which is caused by mutations in the *tem8* gene.

## Gut in *Cmg2*^KO mice is normal under basal conditions, but fails to recover from DSS-induced colitis

To explore the role of CMG2 in intestinal homeostasis, we made use of our previously reported *Cmg2*^KO mice (Bürgi et al, 2017). This mouse line carries a deletion of the exon 3 of *Cmg2*, which encodes a ß-strand within the extracellular von Willebrand A (vWA) domain, and was predicted to prevent proper CMG2 folding and therefore rapid ER-associated proteasomal degradation. Consistently, no shorter CMG2 band could be observed upon transfection of RPE1 cells CMG2Δex3, unless cells were treated with the proteasome inhibitor MG132 (Appendix Fig. S1C). Similarly, we could not detect any lower molecular weight Cmg2 band upon Western blot analysis of the duodenum sample of the *Cmg2*^KO mice, while Cmg2 was readily detectable for control mice (Fig. S1D).

Using this mice model, we found no obvious symptoms associated with intestinal dysfunction (Fig. EV1A), nor abnormalities in crypt architecture detectable by histological examination (Fig. EV1B-D) in *Cmg2*^KO mice. We specifically measured collagen VI, as it has been observed to accumulate over time in both HFS patient nodules and *Cmg2*^KO mouse tissues (Bürgi et al, 2017), due to impaired CMG2-mediated turnover. However, the colon of 8-week-old *Cmg2*^KO mice had normal levels of collagen VI as measured by western blot analysis of protein extracts (Fig. EV1E,F), histological examination (Fig. EV1B), immunofluorescence (Fig. EV1G,H), and qPCR (Fig. EV1I). Additionally, the loss of CMG2 did not affect the number of ISCs and actively proliferating cells, both essential for maintaining epithelial integrity under homeostatic conditions (Sato, 2009; Barker and Clevers, 2010; Barker et al, 2007). Markers for ISCs, including Lgr5 mRNA (Fig. EV1J–L), and indicators of cell proliferation (Fig. EV1M–O) also showed no significant differences between *Cmg2*^KO and *Cmg2*^WT mice, suggesting normal cellular turnover. Furthermore, *Cmg2*^KO mice exhibited no signs of inflammation (Fig. EV1P). Overall, these findings indicate that the loss of CMG2 does not significantly impact the architecture or function of the mouse colon under homeostatic conditions.

To explore this seeming discrepancy between the healthy colons in *Cmg2*^KO mice versus the recurrent, non-infectious diarrhea seen in HFS patients, we examined whether differences between control and KO mice would emerge under stress conditions. We chose to use the well-established dextran sodium sulfate (DSS) model to induce colitis. We first performed a pilot experiment on control mice to determine the adequate DSS concentration to reproducibly trigger the established effects of DSS-induced colitis (Yang and Merlin, 2024; Chassaing et al, 2014; Eichele and Kharbanda, 2017; Kiesler et al, 2015), including body weight loss, diarrhea, rectal bleeding, shortening of colon and increased spleen weight (Appendix Fig. S2A–F). This pilot indicated that 3% DSS was the

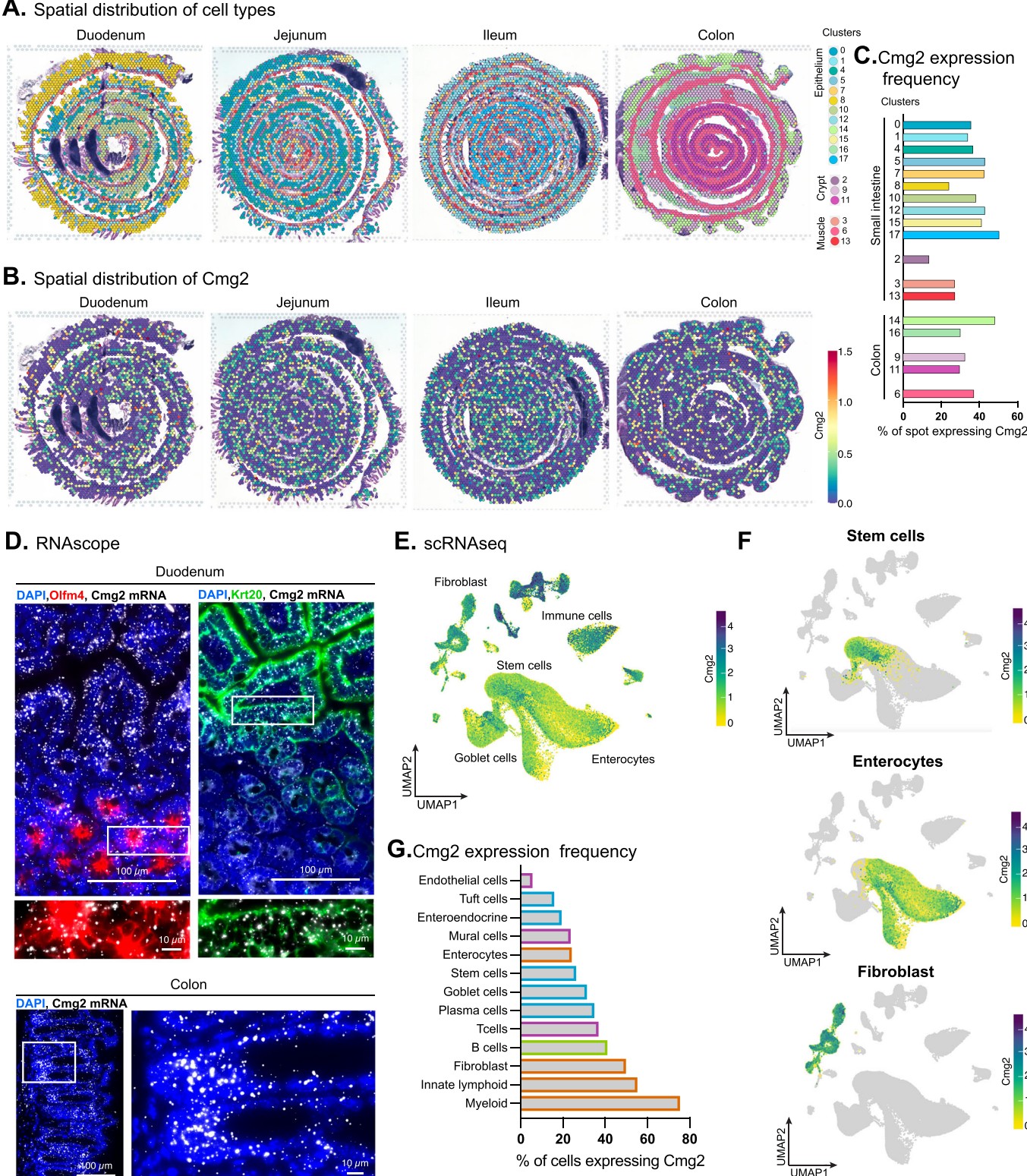

A. Spatial distribution of cell types

B. Spatial distribution of Cmg2

C. Cmg2 expression frequency

D. RNAscope

E. scRNAseq

F. Stem cells / Enterocytes / Fibroblast

G. Cmg2 expression frequency

**Figure 1.  Expression pattern of CMG2 in the gut.**

(A) Clusters of VISIUM spots coupled with tissue H&E staining, replotted from raw data published in Mayassi et al, 2024 (Data ref: Mayassi et al, 2024a). (B) Cmg2 expression on intestinal tissue from VISIUM data, coupled with H&E staining. (C) frequency of VISUM spots expressing Cmg2 among clusters shown in (A). (D) RNAscope in situ hybridization of longitudinal sections of mice duodenum (top panel) or colon (bottom panel) for CMG2 (white) coupled with immunofluorescence staining of Olm4 (red), Keratin20 (green) and DAPI (blue). Representative images of $n = 3$ mice. (E, F) UMAP of scRNAseq replotted from raw data published in Mayassi et al, 2024 (Data ref: Mayassi et al, 2024a). (G) Percentage of cells expressing Cmg2 from scRNAseq (Data ref: Mayassi et al, 2024a).

lowest concentration to reproducibly induce colitis. Therefore, 3% DSS was provided to $Cmg2^{KO}$ mice and their wild-type littermates in the drinking water for 7 days. We daily monitored the disease activity index (DAI) (Yang and Merlin, 2024; Chassaing et al, 2014; Eichele and Kharbanda, 2017) (Fig. 2A,B). Both mouse groups ($Cmg2^{WT}$ and $Cmg2^{KO}$) consumed similar amounts of water (Fig. 2C), leading to similar body weight loss (Fig. 2D), diarrhea and rectal bleeding (Fig. 2E). Until the end of the 7-day DSS treatment, both groups showed a similar, progressive increase in DAI (Fig. 2F) alongside a sharp rise in the intestinal inflammation marker lipocalin-2 (Fig. 2G), confirming the successful induction of colitis. Histological examination immediately after the DSS challenge revealed significant colon alterations in both $Cmg2^{WT}$ and $Cmg2^{KO}$ mice, including loss of crypt architecture (Fig. EV2A), depletion of epithelial cells (Fig. 3A,B), and reduced cell proliferation (Fig. 3A,C). DSS treatment also led to the loss of Lgr5$^+$ stem cells (Fig. 3A,D,E), a common feature of intestinal damage, with similar levels of inflammation observed in both groups (Fig. 3F–I). Overall, these results indicate that CMG2 does not affect the severity of DSS-induced colitis.

After the 7-day DSS treatment, mice were switched to normal drinking water for 3 days, a period known to allow intestinal regeneration in control mice (Yang and Merlin, 2024; Chassaing et al, 2014; Eichele and Kharbanda, 2017) (Fig. 2A). During this phase, $Cmg2^{WT}$ mice showed typical signs of recovery, including a stabilization of body weight (Fig. 2D), cessation of rectal bleeding (Fig. 2E), reduction in DAI (Fig. 2F), and reduced levels of lipocalin-2 in feces (Fig. 2G). In contrast, $Cmg2^{KO}$ mice continued to lose weight (Fig. 2D), ultimately reaching the legal euthanasia limit (> 25% body weight loss), which led to a sudden increase in apparent mortality (Fig. 2H). Moreover, both occult blood and elevated lipocalin-2 levels persisted in the feces of $Cmg2^{KO}$ mice (Fig. 2E–G). Additionally, colons from $Cmg2^{KO}$ mice euthanized on day 10 were significantly shorter than those of their wild-type counterparts (Fig. 2I,J). Together, these findings indicate that while CMG2 does not affect the initial severity of DSS-induced colitis, it plays a crucial role in promoting intestinal recovery after injury, as evidenced by the impaired regeneration observed in $Cmg2^{KO}$ mice.

## Injury-induced colon regeneration is impaired in $Cmg2^{KO}$ mice

To investigate the underlying cellular mechanisms, we performed a detailed analysis of mouse colons after 3 days of DSS withdrawal. In control mice, the crypts were elongated and hypertrophic, a hallmark of the regenerative phase following colitis (Fig. 4A–C) (Yui, 2018; Wang et al, 2019; Erben et al, 2014). In stark contrast, $Cmg2^{KO}$ mice displayed shorter, fewer crypts, and the tissue showed severe alterations. The atrophic crypts in $Cmg2^{KO}$ mice had a squamous morphology, differing from the typical columnar shape

of epithelial cells seen under normal conditions or during regeneration in $Cmg2^{WT}$ mice (Fig. 4A). In addition, the histological inflammation score was significantly higher in $Cmg2^{KO}$ colons during the DSS withdrawal period compared to control mice (Fig. 4D,E; Appendix Fig. S2G–J).

To further examine the damage-induced regeneration process, we assessed epithelial cell proliferation by co-staining for E-cadherin and Ki67 (Fig. 4F). In $Cmg2^{WT}$ mice, colons showed elongated crypts with a higher percentage of E-cadherin + /Ki67+ cells compared to the active disease phase (Day 7), returning to levels similar to those seen under control conditions (Day 0) (Fig. 4F,H). In contrast, $Cmg2^{KO}$ mice exhibited persistently low levels of proliferating epithelial cells from day 7 to 10, with crypts that remained atrophic. Damage in $Cmg2^{KO}$ colons appeared to worsen even after DSS removal, as indicated by a further reduction in epithelial cell numbers (Fig. 4G). In addition, these colons had more infiltrating neutrophils than those of WT mice (Fig. 4I) and showed increased expression of the inflammation marker lipocalin-2 (Fig. 4J). Upon DSS challenge and intestinal damage, increased expression of collagen, and especially Collagen VI, has been observed and shown to be essential for intestinal recovery (Molon et al, 2023). Because Cmg2 is involved in Collagen VI homeostasis, we investigated whether the colon of DSS-treated $Cmg2^{KO}$ mice showed increased levels of Collagen VI compared to their WT littermates. Consistent with the literature (Molon et al, 2023), WT mice exhibited a marked increase in Collagen VI expression during the acute phase of colitis, with levels returning toward baseline following DSS withdrawal (Fig. EV2A). A similar expression pattern was observed for $Cmg2^{KO}$ mice, with no significant differences in $Col6a1$ mRNA levels between WT and KO animals throughout the time course of the experiment. This observation was further confirmed at the protein level by western blot (Fig. EV2D,E) and immunohistochemistry analyses (Fig. EV2B,C), suggesting that the impaired regenerative capacity observed in $Cmg2^{KO}$ mice is not caused by Collagen VI accumulation.

Collectively, these findings indicate that $Cmg2^{KO}$ mice fail to transition from the active disease phase to the regenerative phase after DSS withdrawal, leading to increased damage and impaired repair.

## CMG2 is dispensable for YAP/TAZ-mediated reprogramming to fetal-like stem cells

Since $Cmg2^{KO}$ mice fail to regenerate their gut upon DSS challenge, we next examined which key steps of the fetal-like regenerative response (Yui, 2018; Murata et al, 2020; Nusse et al, 2018) might be affected by Cmg2 depletion. First, we tested whether the expression of Ly6a, a marker of fetal-like stem cells, was influenced by CMG2 in DSS-injured colons. As expected, Ly6a mRNA levels were low under homeostatic conditions and increased significantly 3 days

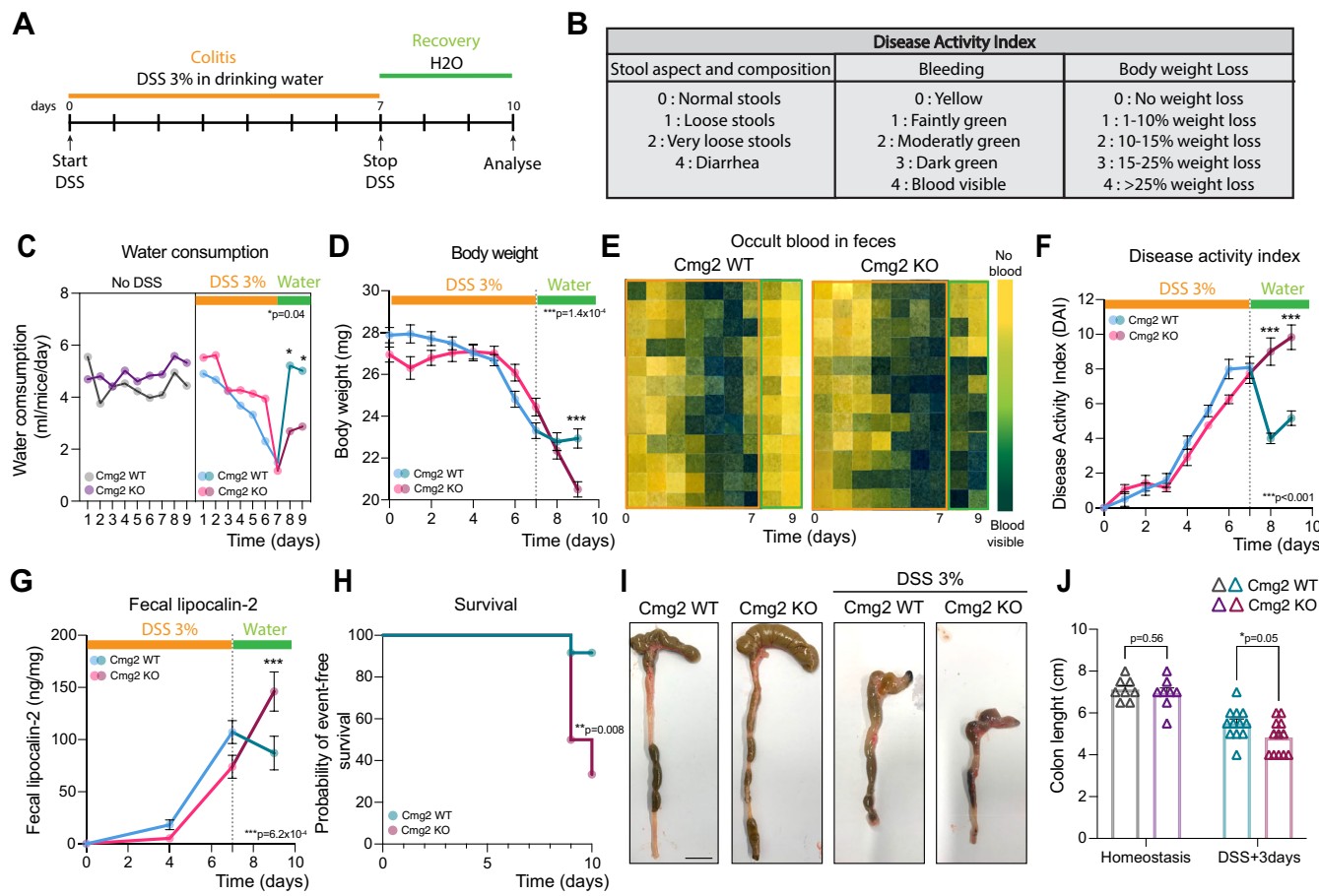

**Figure 2. *Cmg2*[KO] mice fail to recover from DSS-induced colitis.**

Eight-week-old *Cmg2*[WT] or *Cmg2*[KO] males were given 3% Dextran-Sulfate-Sodium (DSS) in their drinking water for 7 days, then switched to regular drinking water for 3 days to recover. (A) Experimental scheme of DSS-induced colitis. (B) Disease activity index (DAI) scoring and (C) water consumption were measured daily during the 10-day experiment. In (C), the results are median. Each dot represents the mean of $n = 2$ or 3 different cages. *P* values were obtained using a two-way ANOVA Tukey's multiple comparisons test. (D) Body weight loss, (E) the aspect of the feces and presence of occult blood were monitored and used to evaluate the Disease activity index in (F). (D–F) The results are mean ± SEM. Each dot represents the mean of $n = 12$ mice per genotype. *P* values were obtained by two-way ANOVA Šídák's multiple comparisons test. (F) Exact *P* value for days 8 and 9 are, respectively, $P = 4.2 \times 10^{-13}$ and $P = 1.2 \times 10^{-11}$. In (E), the presence of blood in feces was detected using a urine strip test from collected feces. Each row corresponds to a single mouse and each column to a different timepoint. Feces were collected daily, and (G) fecal lipocalin-2 was quantified by ELISA. Results are mean ± SEM. Each dot represents the mean of $n = 12$ mice per genotype. *P* values were obtained using a two-way ANOVA Šídák's multiple comparisons test. Excessive body weight loss ( > 25%) required animal euthanasia, as shown in (H) probability of survival. On day 10, mice were euthanatized, organ collected and (I, J) colon length were measured. Representative images of $n = 8$ mice per genotype for untreated mice and $n = 12$ mice per genotype for DSS-treated mice. Scale bar, 1 cm. (J) Results are mean ± SEM. Each symbol represents a single measurement. *P* values were obtained using an unpaired *t* test.

after DSS withdrawal in control mice. (Fig. 5A). Interestingly, this same pattern was observed in *Cmg2*[KO] mice (Fig. 5A). Immuno-fluorescence staining further confirmed the emergence of a fetal-like stem cell population in the colons of both control and *Cmg2*[KO] mice during regeneration (Fig. 5B,C). Given that this switch to fetal-like stem cells is heavily dependent on YAP/TAZ signaling pathways (Yui, 2018; Sprangers et al, 2021), we investigated whether CMG2 loss affected YAP/TAZ signaling during intestinal regeneration. In addition to the fetal-like stem cell marker Ly6a, which is a YAP/TAZ target gene, we measured the mRNA levels of two other YAP target genes, Cyr61 and CTGF (Fig. 5D,E), both of which were significantly upregulated in the injured colons of *Cmg2*[KO] mice compared to DSS-injured *Cmg2*[WT] mice (Fig. 5D,E). These findings suggest that the absence of CMG2 does not impair

YAP/TAZ-dependent reprogramming of intestinal cells into fetal-like stem cells.

## CMG2 is critical for restoring the Lgr5+ intestinal stem cell pool

We then investigated whether the fetal-like stem cells observed in *Cmg2*[WT] and *Cmg2*[KO] mice could transition into adult Lgr5+ ISCs for the replenishment of Lgr5+ stem cells, which is crucial to drive epithelial repair (Yui, 2018; Murata et al, 2020; Nusse et al, 2018). After 3 days of DSS withdrawal, Lgr5 expression, monitored by RNAscope, returned to homeostatic levels in the colons of *Cmg2*[WT] mice, as expected (Fig. 5F,G). In marked contrast, Lgr5 expression remained extremely low in *Cmg2*[KO] mice, comparable to levels seen

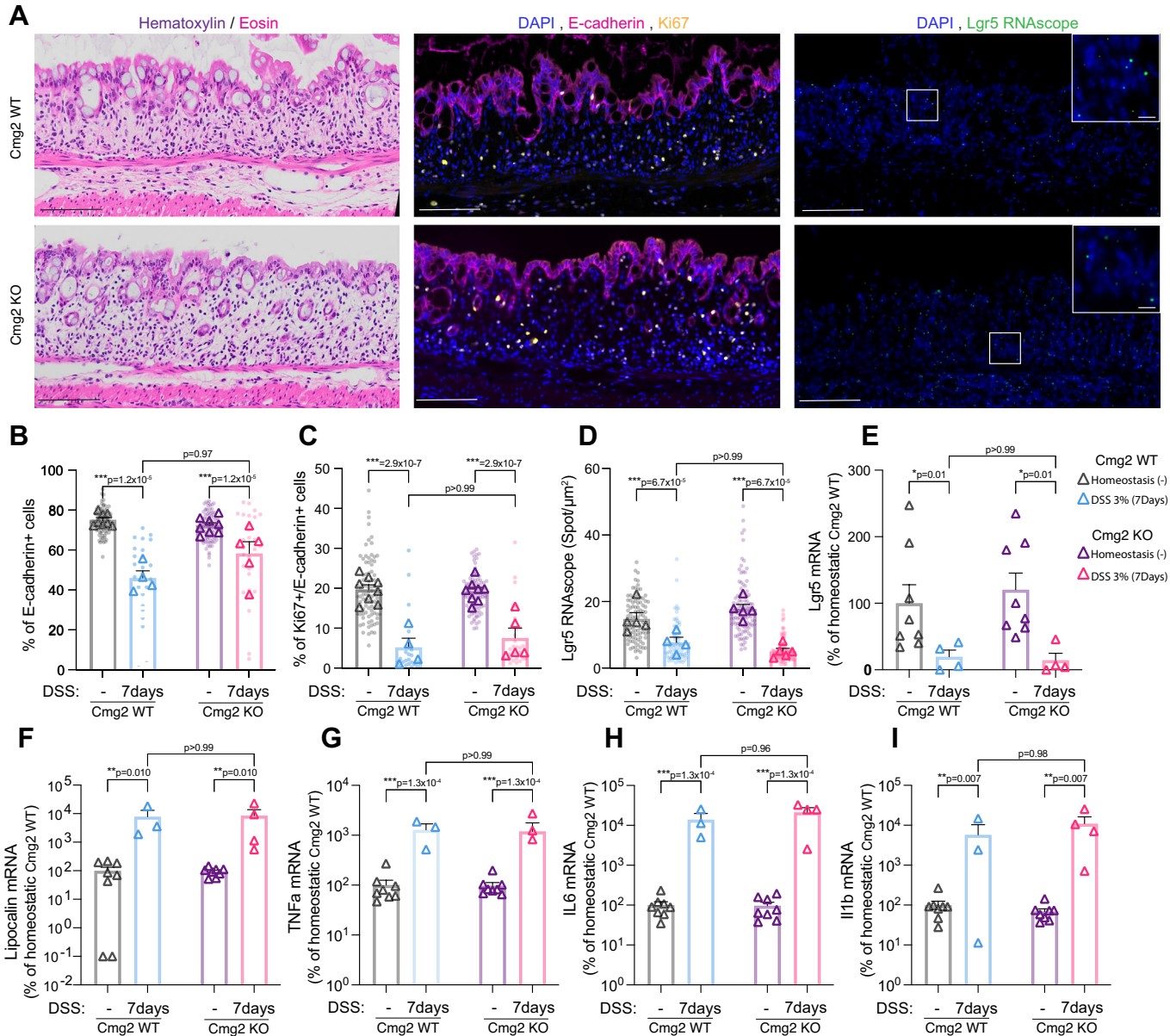

**Figure 3. CMG2 loss does not enhance the severity of DSS-induced colitis.**

Eight-week-old *Cmg2*^WT or *Cmg2*^KO males were given 3% Dextran-Sulfate-Sodium (DSS) in their drinking water for 7 days. On day 7, mice were euthanized, and the colon was collected. (A) Colon tissues were stained for Hematoxylin/Eosin (left panel), stained with anti-ki67, anti-VE-cadherin and DAPI (middle panel) or stained for RNAscope in situ hybridization against Lgr5 (right panel). Representative images of at least $n = 4$ mice per genotype. Scale bar,100 μm for main image and 10 μm for magnification. (B) % of E-cadherin+ cells, (C) % of E-cadherin + /Ki67+ cells, and (D) number of Lgr5 mRNA spots per μm², were quantified. Results are mean ± SEM. Each dot represents a single measurement, and each triangle represents the mean per mouse. At least $n = 4$ mice per genotype were quantified. *P* values obtained by two-way ANOVA Šídák's multiple comparisons test. (E-I) Quantitative real-time PCR analysis of colonic tissues from *Cmg2*^WT and *Cmg2*^KO for (E) Lgr5, (F) Lipocalin-2 and (G-I) inflammatory cytokine genes. Results are mean ± SEM. Each symbol represents a single measurement. At least $n = 3$ mice per genotype were quantified. *P* values obtained by two-way ANOVA Šídák's multiple comparisons test.

during the active disease phase (7 days of DSS treatment) (Fig. 5F,G). qPCR analysis reinforced these findings, showing significantly reduced levels of Lgr5 and three other ISC marker genes (Axin2, Ascl2, and Cyclin D1), all of which are Wnt targets, in *Cmg2*^KO mice compared to their wild-type littermates (Fig. 5H). These results indicate that, despite the presence of Ly6a+ fetal-like stem cells, the conversion into Lgr5+ ISCs failed to occur in the

absence of CMG2, suggesting that CMG2 expression is necessary to allow the transition of these Ly6a+ cells into adult ISC. To verify this hypothesis, we leveraged an existing spatial transcriptomics dataset in which mice received 2.5% DSS in drinking water for 5 days and were allowed to recover up to 72 days (Fig. EV3A). Consistent with the data in Fig. 1, *Cmg2* was broadly expressed across all clusters under homeostatic conditions. As expected, gene

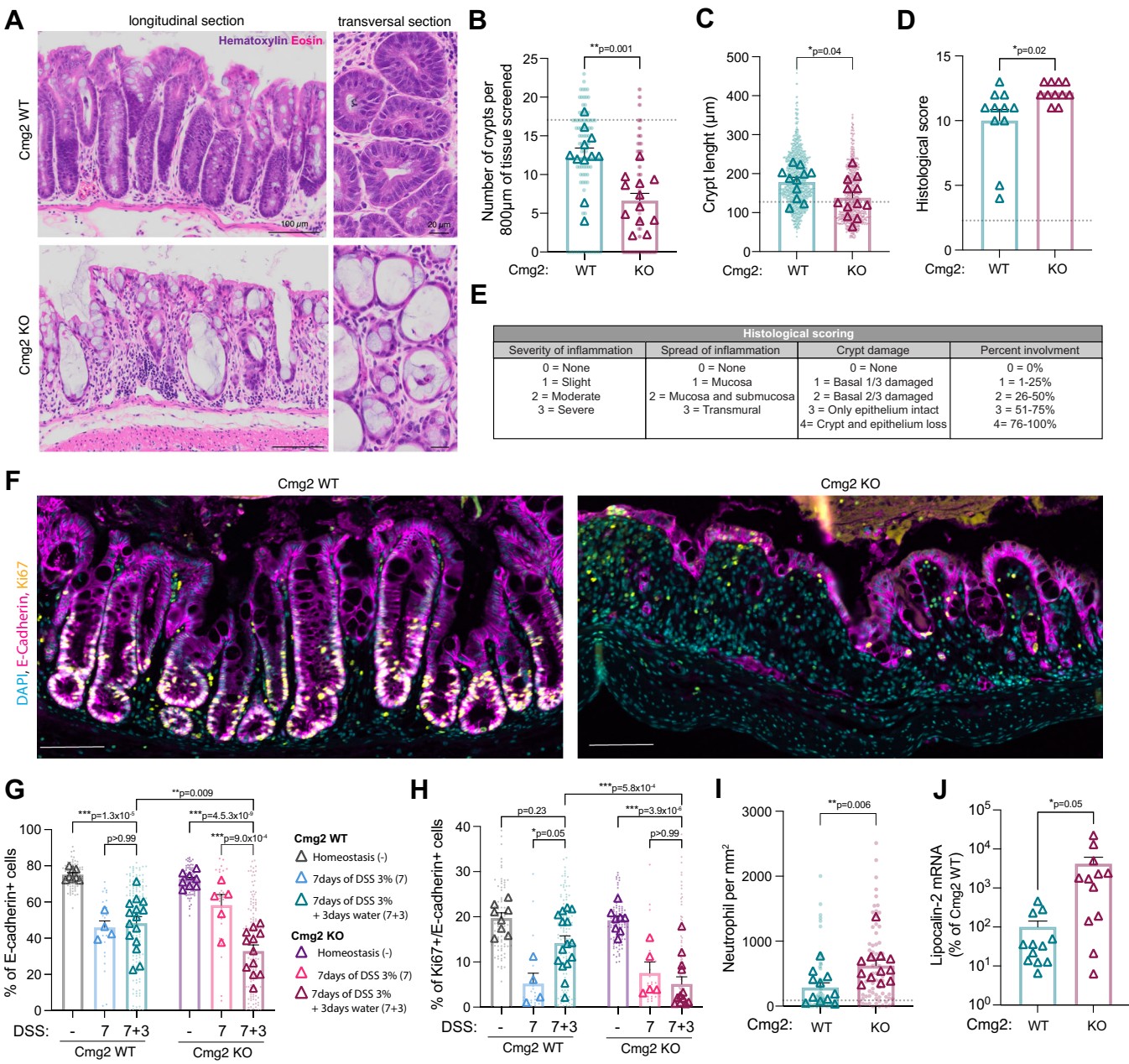

**Figure 4. Injury-induced gut regeneration is impaired in *Cmg2*[KO] mice.**

8-week-old *Cmg2*[WT] or *Cmg2*[KO] males were given 3% Dextran-Sulfate-Sodium (DSS) in their drinking water for 7 days, then switched to regular drinking water for 3 days to recover. On day 10, mice were euthanized, and their colons were collected. (A) Longitudinal (left panel) and transversal section (right panel) of colon tissues were stained for Hematoxylin/Eosin. Representative images of at least $n = 11$ mice per genotype. Scale bar,100 μm. (B) The number of crypts per 800 μm of tissue screened, (C) crypt length and (D, E) histological scores were quantified. Results are mean ± SEM. Each dot represents a single measurement, and each triangle represents the mean per mouse. The dotted line represents homeostatic levels of *Cmg2*[WT] mice. At least $n = 10$ mice per genotype were quantified. *P* values obtained by two-tailed unpaired *t* test. (F) Colonic sections were stained with anti-ki67, anti-VE-cadherin, and DAPI. Representative images of at least $n = 10$ mice per genotype. Scale bar, 100 μm. (G) % of E-cadherin + , and (H) % of E-cadherin + /Ki67+ cells at homeostasis (−), after 7 days of DSS (7) or 3 days after DSS withdrawal (7 + 3) were quantified. Results are mean ± SEM. Each dot represents a single measurement, and each triangle represents the mean per mouse. At least $n = 4$ mice per genotype were quantified. *P* values obtained by two-way ANOVA Šídák's multiple comparisons test. (I) Immunostaining of neutrophil was performed on a colonic section using S100A9 antibody, and the number of neutrophils per mm² quantified. Results are mean ± SEM. Each dot represents a single measurement, and each triangle represents the mean per mouse. At least $n = 11$ mice per genotype were quantified. *P* values obtained by two-tailed unpaired *t* test. (J) qPCR analysis of colonic tissues from *Cmg2*[WT] and *Cmg2*[KO] for the intestinal inflammation marker Lipocalin-2. Results are mean ± SEM. $n = 12$ mice per genotype were quantified. *P* values obtained by two-tailed unpaired *t* test.

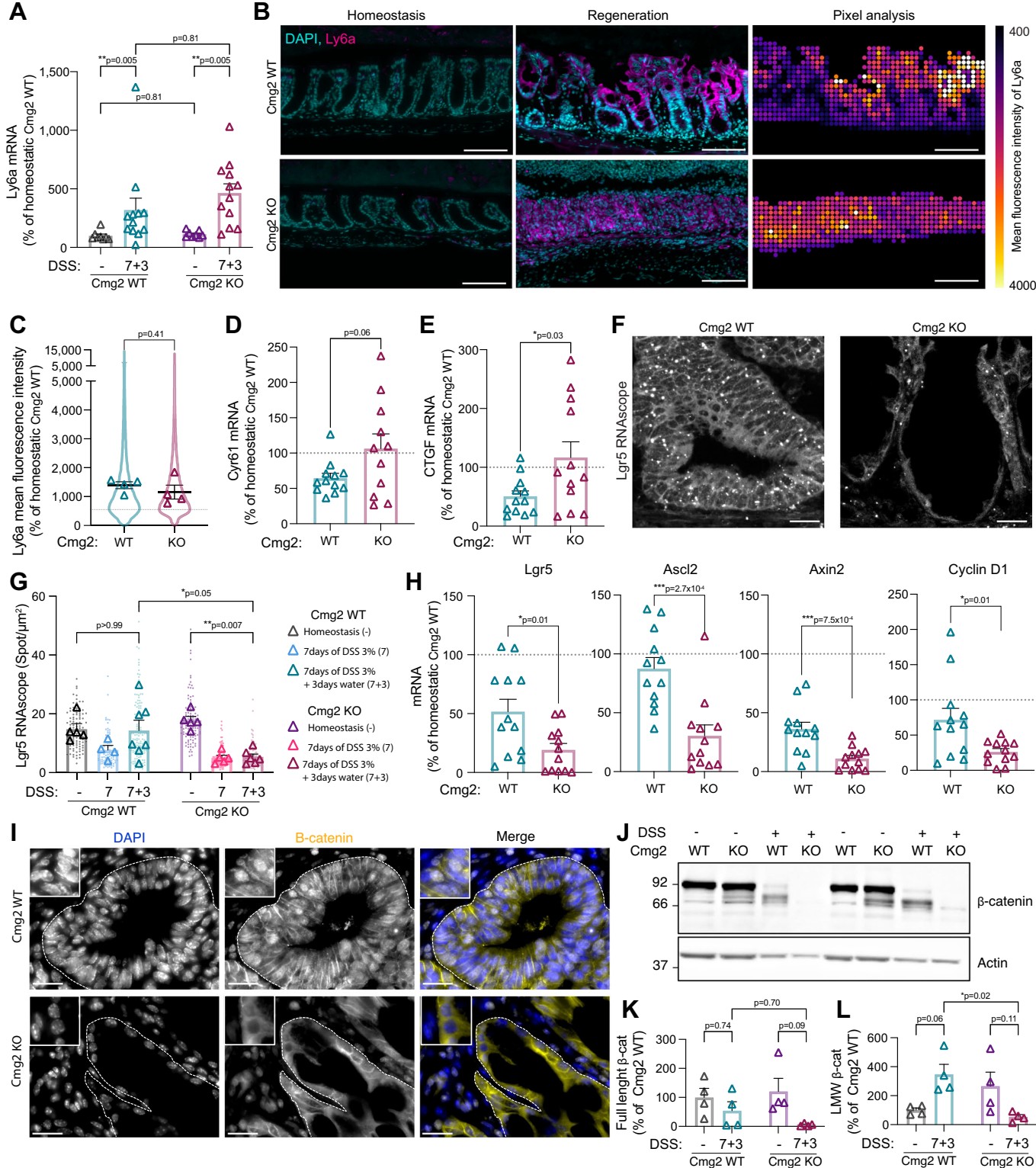

**Figure 5. CMG2 is dispensable for YAP/TAZ-dependent reprogramming into fetal-like stem cells but essential for the replenishment of the Lgr5[+] ISC pool.**

(A) qPCR analysis of colonic tissues from $Cmg2^{WT}$ and $Cmg2^{KO}$ during homeostasis (−) or 3 days after DSS withdrawal (7 + 3) for Ly6a. Results are mean ± SEM. Each symbol represents the mean per mouse. At least $n = 10$ mice per genotype were quantified. $P$ values obtained by two-way ANOVA Šídák's multiple comparisons test. (B, left and middle panel) Colonic sections were stained with anti-Ly6a, and DAPI and (B, right panel) coarse-grain analysis was performed. Representative images of $n = 4$ mice per genotype. Scale bar, 100 μm. (C) Results are presented as a violin plot of the Ly6a mean intensity of all data points from the coarse-grain analysis. Each symbol represents the mean per mouse of $n = 4$ mice per condition. Results are mean ± SEM. The dotted line represents the average homeostatic levels of $Cmg2^{WT}$. $P$ values were obtained by two-tailed unpaired $t$ test. (D, E) qPCR analysis of colonic tissues from $Cmg2^{WT}$ and $Cmg2^{KO}$ 3 days after DSS withdrawal for YAP target genes (D) Cyr61 and (E) CTGF. Results are mean ± SEM. At least $n = 11$ mice per genotype were quantified. $P$ values obtained by two-tailed unpaired $t$ test. The dotted line represents the average homeostatic levels of $Cmg2^{WT}$. (F, G) Colonic sections were stained for RNAscope in situ hybridization against Lgr5. Representative images of at least $n = 4$ mice. Scale bar, 100 μm. (G) Lgr5 mRNA spots per μm$^2$ at homeostasis (−), after 7 days of DSS (7) or 3 days after DSS withdrawal (7 + 3) were quantified. Results are mean ± SEM. At least $n = 4$ mice per genotype were quantified. $P$ values obtained by two-way ANOVA test. (H) qPCR analysis of colonic tissues from $Cmg2^{WT}$ and $Cmg2^{KO}$ 3 days after DSS withdrawal for ISC genes Lgr5, Ascl2, Axin2 and Cyclin D1. Results are mean ± SEM. At least $n = 12$ mice per genotype were quantified. $P$ values obtained by two-tailed unpaired $t$ test. The dotted line represents the mean of homeostatic $Cmg2^{WT}$ mice. (I) Colonic sections were stained with anti-β-catenin and DAPI. Representative images of $n = 4$ mice per genotype. Scale bar, 100 μm. The dotted line represents the average homeostatic levels of $Cmg2^{WT}$. (J) Colon lysates were western blotted against β-catenin and Actin. (K) Full-length (95 kDa) or (L) low-molecular-weight (LMW) β-catenin/loading control ratio was quantified and normalized to the mean of $Cmg2^{WT}$. Results are mean ± SEM, and $n = 4$ mice per genotype were quantified. $P$ values obtained by two-way ANOVA Tukey's multiple comparisons test.

expression was highly perturbed at day 12 with the emergence of inflammation/regeneration clusters (Fig. EV3B). In particular, cluster 8 which is enriched with the fetal-like marker Ly6a (Fig. EV3C) also showed $Cmg2$ expression (Fig. EV3D), corroborating our hypothesis that CMG2 might be cell-autonomously required to achieve fetal-like to ISC transition. Interestingly, we observed that the progressive return to the steady-state transcriptional landscape between day 30 and 72 was accompanied with an increased $Cmg2$ expression both at the expression level (2.5-fold increase) and percentage of cells expressing $Cmg2$ (2.2-fold increase) (Fig. EV3A,D). This raises the interesting possibility that CMG2 plays a context-dependent role in Wnt signaling not only because of the tissue, but also because of the history or the present structure of the tissue.

Following the discovery of the reduced levels of Wnt targets, we next investigated the role of CMG2 in modulating Wnt signaling following intestinal damage. Stemness in the intestinal epithelium relies on Wnt signaling (Meyer et al, 2022; Gehart and Clevers, 2019), which is tightly regulated by various signaling pathways (Barry, 2013; Cai et al, 2010; Gregorieff et al, 2015; Murata et al, 2020; Walter et al, 2022; Deng et al, 2018), and crucial for effective injury-induced intestinal repair (Guillermin et al, 2021). Initially, we quantified the expression of key molecular components involved in Wnt signaling in the mouse colon 3 days after DSS withdrawal using qPCR. The mRNA levels of the Frizzled co-receptor LRP6, β-catenin (Fig. EV2F,G), and Wnt ligands (Wnt5a, 5b, and 2b) were comparable between the colons of $Cmg2^{WT}$ and $Cmg2^{KO}$ mice (Fig. EV2H), suggesting that CMG2 does not significantly influence the abundance of these major Wnt signaling players. Interestingly, the spatial transcriptomic analysis mentioned above revealed that the expression levels of $Cmg2$ and $Lrp6$ are correlated, with both genes showing increased expression during the return to steady-state following DSS-induced injury (Fig. EV3D,E). This correlation is consistent with a possible functional relationship between CMG2 and LRP6 during intestinal repair.

Next, we analyzed β-catenin activation in the colon of Cmg2$^{WT}$ and Cmg2$^{KO}$ mice during the recovery phase. Strikingly, while β-catenin was prominently detected in the nuclei of hypertrophic crypts in $Cmg2^{WT}$ mice through immunostaining, the nuclei of cells in the atrophic crypts of $Cmg2^{KO}$ mice were remarkably devoid of β-catenin (Fig. 5I). Western blot analysis revealed that β-catenin

underwent cleavage in the colon of $Cmg2^{WT}$ mice 3 days after DSS withdrawal (Fig. 5J–L), a hallmark of enhanced transcriptional activation (Goretsky et al, 2018). In contrast, $Cmg2^{KO}$ mice exhibited extremely low levels of both full-length and cleaved β-catenin during DSS treatment (Fig. 5J–L), despite displaying normal levels under basal conditions (Fig. EV2I,J).

These results are in apparent discrepancy with the study on human duodenal organoids derived from a Hyaline Fibromatosis Syndrome patient, which proposed that the loss of CMG2 expression/function does not affect the cells of the epithelial barrier (van Rijn et al, 2020). These human organoids, like most others, were generated and maintained in very high Wnt-containing media. Under these culture conditions, the CMG2–LRP6–Wnt axis might not be necessary. Also, the HFS-patient-derived duodenoids were not entirely normal, showing intercellular blisters indicative of some defect in epithelial structure/function. Future studies using different organoid culture media will be necessary to elucidate this point.

Based on these collective findings, we propose that CMG2 plays an essential role in the reactivation of Wnt signaling, facilitating the transition from fetal-like to adult ISCs necessary for effective intestinal repair.

# Discussion

The present study underscores a critical and previously unknown role of CMG2 in gut function. The most severe form of HFS is often fatal due to intractable diarrhea. Our results show that $Cmg2^{KO}$ mice exhibited no abnormalities in the colon under normal conditions, correlating with normal function of enterocytes derived from HFS patient cells (van Rijn et al, 2020). Differences became apparent when the mice were subjected to chemically induced colitis. While both control and $Cmg2^{KO}$ mice showed similar disease severity upon DSS treatment, the knockout mice failed to regenerate their colons, resulting in persistent bloody diarrhea and significant weight loss, closely mirroring the symptoms seen in HFS patients.

One could envision that Cmg2KO mice have a defect in peristalsis, resulting in longer dwell times and possibly a higher effective dose of DSS to the KO epithelium. We, however, did not observe any signs of

intestinal obstruction or fecal retention in Cmg2KO mice. Animals were single-caged for 30 min to collect feces. We did not observe any difference in amounts collected from WT and KO mice, arguing against a substantial difference in transit time of gut contents. Moreover, if DSS affected the recovery, one would have expected a more severe histological phenotype in the colon of Cmg2KO since the tissue likely already attempts regeneration during the 7 days of DSS treatment. But this was not the case. Therefore, while we cannot formally rule out the presence of residual DSS in Cmg2KO mice during the DSS withdrawal phase, there is currently no indication that this was the case.

Our work demonstrates the implication of Cmg2 in intestinal regeneration after damage, a process that is thought to occur in at least two phases. Cmg2 was found to be non-essential during the initial dedifferentiation phase, characterized by high YAP activation, with repression of Wnt signaling and ISC gene expression (Barry, 2013; Cai et al, 2010; Gregorieff et al, 2015), since we observed that fetal-like stem cells were present in the damaged colon of both *Cmg2*^WT and *Cmg2*^KO mice. Our data indicate that Cmg2 is essential for the second phase to occur, with Wnt- and Ascl2-dependent reacquisition of Lgr5+ stemness (Ouladan and Gregorieff, 2021), and the subsequent restoration of intestinal homeostasis after injury. The fact that Cmg2 is not required for basal gut homeostasis but is essential for replenishing the ISC pool to occur aligns with the idea that Lgr5+ cells are dispensable for maintaining normal epithelial function (Tian, 2011; Asfaha et al, 2015). During intestinal regeneration, the transition of these fetal-like stem cells to adult ISCs, require the downregulation of YAP and reactivation of Wnt signaling (Schuijers et al, 2015; Lustig et al, 2002; Carmon et al, 2011; de Lau et al, 2011; Oost et al, 2023; Viragova et al, 2024), suggesting a major role of Cmg2 in tuning these signaling pathways.

Our findings expand the growing network of connections between Cmg2 and Wnt signaling pathways, as well as the connection to stem cells. Previous work on CMG2 identified it as an interactor of Wnt signaling co-receptor LRP6 through studies on anthrax toxin receptors (Lustig et al, 2002). We confirmed this interaction, demonstrating that LRP6 and CMG2 co-internalize during toxin entry (Abrami et al, 2008). CMG2 has also been shown to modulate LRP6 levels in a concentration-dependent manner, where both low and high levels of CMG2 reduce LRP6 abundance in cultured cells (Abrami et al, 2008). In mouse 3T3-L1 cells, downregulation of Cmg2 consistently impaired Wnt-induced β-catenin stabilization (Abrami et al, 2008). CMG2 not only connects to canonical LPR6-dependent Wnt signaling. We indeed found that during zebrafish embryonic development, CMG2 is necessary for the non-canonical Wnt-dependent oriented cell division of epiblast cells, necessary for embryo extension (Castanon et al, 2020, 2013), a process that still involved LRP6. Recent studies in cancer biology extended the connection of CMG2-Wnt to stem cells (Ji et al, 2018). CMG2 levels correlated with increased expression of stemness-related genes, such as Lgr5, as well as enhanced self-renewal and metastatic potential (Ji et al, 2018). Manipulating CMG2 levels through over-expression or silencing altered nuclear β-catenin, consistent with previous observations in 3T3-L1 cells and our current findings that Cmg2 deficiency impaired β-catenin activation and Lgr5+ ISC in the damaged. Noteworthily, the absence of Cmg2 does not result in the same effects as LRP6 or Wnt deficiencies, which can cause early lethality or milder phenotypes (Vanamerongen and Berns, 2006). This indicates that Cmg2 modulates Wnt signaling in a context-specific manner within the gut.

In conclusion, our study demonstrates that CMG2 is a key player in gut regeneration following injury, specifically impacting the Wnt-mediated replenishment of the Lgr5+ stem cell pool. Combined with its established role in gastric stem cells, this work suggests a broader function for CMG2 in stem cell biology, particularly in tissue regeneration involving fetal-like reversion, as also seen in the stomach (Viragova et al, 2024; Willet et al, 2018; Brown et al, 2022). Future studies will be required to reach a mechanistic understanding of the context-dependent impact of CMG2 on Wnt signaling. From a clinical perspective, our findings provide a molecular basis for the persistent diarrhea seen in HFS patients. In HFS infants, a diarrheal event may lead to failure in replenishing the stem cell pool, thus preventing reestablishment of the epithelial barrier. These insights suggest that HFS patients might benefit from therapeutic approaches currently used for chronic diarrhea conditions, such as inflammatory bowel diseases (IBD).

## Methods

**Reagents and tools table**

| Reagent/resource | Reference or source | Identifier or catalog number |
| --- | --- | --- |
| **Experimental models** | | |
| Cmg2^KO (*M. musculus*) | Bürgi et al, 2017 | N/A |
| hTERT RPE-1 (*H. sapiens*) | ATCC | CRL-4000 |
| **Recombinant DNA** | | |
| pcDNA3.5-Cmg2-V5-HIS | This study | N/A |
| **Antibodies** | | |
| Mouse anti-Actin | Millipore | MAB1501 |
| Rabbit anti-CollagenVI | Abcam | ab6588 |
| Mouse anti-E-cadherin | Abcam | ab76055 |
| Rabbit anti-Ki67 | Abcam | ab16667 |
| Rabbit anti-S100A9 | Novus Biological | NB110-89726 |
| Rabbit anti-Bcatenin | Sigma | C2206 |
| Rabbit anti-Bcatenin | BD biosciences | 610154 |
| Rat anti-Ly6a | Abcam | ab51317 |
| Mouse anti-V5-HRP | Invitrogen | R960-25 |
| Rabbit anti-Olfm4 | Cell signaling | 39141 |
| Rabbit anti-KRT20 | Cell signaling | 13063 |
| Rat anti-Cmg2 | Hybridomas Supernatant | N/A |
| Goat anti-Cmg2 | R&D Systems | AF2940 |
| Mouse anti-GAPDH | Invitrogen | 398600 |
| Donkey anti-Mouse IgG-HRP | GE Healthcare | NA931V |
| Donkey anti-Rabbit IgG-HRP | GE Healthcare | NA934V |
| Goat anti-Mouse IgG-Alexa488 | ThermoFisher Scientific | A-11029 |
| Donkey anti-Mouse IgG-Alexa568 | ThermoFisher Scientific | A-10037 |

| Reagent/resource | Reference or source | Identifier or catalog number |
|---|---|---|
| Donkey anti-Mouse IgG-Alexa647 | ThermoFisher Scientific | A-31571 |
| Donkey anti-Rabbit IgG-Alexa488 | ThermoFisher Scientific | A-21206 |
| Donkey anti-Rabbit IgG-Alexa568 | ThermoFisher Scientific | A-10042 |
| Donkey anti-Rabbit IgG-Alexa647 | ThermoFisher Scientific | A-31573 |
| Goat anti-Rat IgG-Alexa568 | ThermoFisher Scientific | A-11077 |
| DAPI | ThermoFisher Scientific | D1306 |
| Hoechst | Sigma | 94403 |
| **Oligonucleotides and other sequence-based reagents** | | |
| PCRprimer | This study | Table EV1 |
| **Chemicals, enzymes, and other reagents** | | |
| Dextran sulfate sodium | MP Biomedicals | 160110 |
| Multistix 8SG | Siemens | N/A |
| Prolong Gold Antifade Mounting medium | ThermoFisher Scientific | P36930 |
| RNAscope Multiplex Fluorescent V2 assay | Bio-techne | 323110 |
| Mm-Lgr5 | Bio-techne | 312171 |
| Mm-Antxr2 | Bio-techne | 468651 |
| Mm-Ppib | Bio-techne | 313911 |
| Mm-DapB | Bio-techne | 310043 |
| TSA Opal570 | Akoya Biosciences | FP1488001KT |
| Duoset murine Lcn-2 ELISA kit | R&D Systems | DY1857 |
| RNA easy mini extraction kit | Qiagen | 74104 |
| SYBR MasterMix | Life Technology | A25742 |
| Dulbecco's Modified Eagle Medium | Sigma Life Science | D5030 |
| In-Fusion® Snap Assembly Master Mix | Takara | 638948 |
| **Software** | | |
| GraphPad Prism v10 | https://www.graphpad.com | N/A |
| QuPath | Bankhead et al, 2017 | N/A |
| STARDIST Qupath extension | https://github.com/stardist/stardist | N/A |
| ImageJ | https://imagej.nih.gov/ij/index.html | N/A |
| **Other** | | |
| VS200 whole slide scanner | Olympus | N/A |

## Animals

CMG2 knockout mice were generated by targeted deletion of the gene exon 3 as described previously (Bürgi et al, 2017). Cmg2 mutant mice were generated and backcrossed for at least 10 generations onto the C57Bl6/J genetic background (Charles River Laboratories). $Cmg2^{HET}$ mice were kept in a specific pathogen free (SPF) environment with 12 h light and 12 h dark cycle with ad libitum access to food and water. Experiments were performed using littermate from the different genotypes and animals housed depending on their genotype. During the conduct of DSS experiment and during all data analysis, mice genotype was blinded. Characteristics of animals used in this study as well as animal care and monitoring during DSS experiment are described in the Supplementary information. For animal experimentation, all procedures were performed according to protocols approved by the Veterinary Authorities of the Canton Vaud and according to the Swiss Law (License VD 3497, EPFL) and were in accordance with the ARRIVE guidelines and 3R principle (Reduction, Replacement, Refinement) for laboratory animals.

## DSS-induced colitis

Eight-week-old $Cmg2^{WT}$ or $Cmg2^{KO}$ males were given 3% Dextran Sulfate Sodium (MP Biomedicals:160110) in the drinking water for 7 days, then switched to regular drinking water for 3 days and allowed to recover. Control 8-week-old $Cmg2^{WT}$ or $Cmg2^{KO}$ males were given drinking water that did not contain DSS. Water consumption was similar in DSS-treated or control mice. During DSS treatment and recovery, mice were weighed, scored daily and euthanized if they developed severe disease based on the scoring criteria described in Supplementary information (weight loss, coat appearance, eyes/nose discharge, breathing, activity and posture). Mice were housed in individual cage for 30–60 min daily in order to collect mice feces. Feces were either stored at -20° for further analyses or diluted at 100 mg/ml in PBS 0,1% Tween-20 and used to evaluate the presence of occult blood using urine test strip (Multistix 8SG). Daily evaluation of Disease activity Index (DAI) (described in Supplementary information) was performed, and based on stool consistency, presence of occult blood and body weight loss. On day 10, mice were euthanized with an injection of pentobarbital. Colon was collected, measured and weighted. Tissues were divided into two parts, the first part being stored at −80 °C for RNA and protein extraction and the second part fixed in 4% paraformaldehyde for histological studies.

## Histological studies

Tissues collected after euthanasia were fixed in 4% paraformaldehyde overnight at 4 °C, embedded in paraffin and sectioned at 4 µm thickness. Sections were deparaffinized with xylene and rehydrated with distilled water through a series of graded alcohol. Tissues were stained with hematoxylin and eosin (H&E), Sirius Red or Alcian blue, using standard protocols. Each colon was scored by a pathologist who assigned four scores based on severity and spread of inflammation, crypt damage and percent involvement. For immunohistochemistry, de-waxed samples were subjected to antigen retrieval by boiling in citrate buffer (10 mM, pH 6.0) for 20 min, except for Sca1/Ly6a immunostaining, which was subjected to antigen retrieval by treating with Proteinase K (20 µg/ml) for 10 min at room temperature. Non-specific antigenic sites were blocked with 1% bovine serum albumin (BSA), and samples were hybridized with primary antibody in a humidity chamber at 4 °C overnight. After washing, samples were incubated with secondary antibodies for 1 h at room temperature. After washing, tissues were counterstained with DAPI (1/5000 dilution) and mounted with Prolong

Gold Antifade Mounting medium (ThermoFisher, P36930). Primary antibodies used were: anti-actin (dilution 1/400), anti-CollagenVI (dilution 1/1000), anti-Ki67 (dilution 1/1000) anti-E-cadherin (dilution 1/500), anti-Bcatenin (dilution 1/200), anti-S100A9 (dilution 1/200) (dilution 1/200), and anti-sca1 (dilution 1/200). Secondary antibodies were all used at 1/1000 dilution.

## RNAscope

RNAscope Multiplex Fluorescent V2 assay (Bio-techne, Cat. No. 323110) was performed according to the manufacturer's protocol on 4-μm paraffin sections with 15 min target retrieval at 95 °C and Protease III 30 min at 40 °C, hybridized with the probes Mm-Lgr5-C3 (Bio-techne, Cat. No. 312171-C3), Mm-Antxr2-C1 (Bio-techne, Cat No. 468651) Mm-Ppib-C1 (Bio-techne, Cat. No. 313911) as positive control and DapB-C1 (Bio-techne, Cat. No. 310043) as negative control at 40 °C for 2 h and revealed with TSA Opal570 (Akoya Biosciences, Cat. No. FP1488001KT). Tissues were counter-stained with Hoechst (dilution 1/10000), antii-krt20 (dilution 1/600), and anti-olfm4 (dilution 1/250) and mounted with Prolong Gold Antifade Mounting medium (ThermoFisher, P36930).

## Image acquisition and analysis

Brightfield and fluorescent images were acquired on an Olympus VS200 whole slide scanner using OlyVIA software and ×20 air objective or ×40 air objective. Detailed information about technical specifications is in the Supplementary information.

For all analyses, annotations areas were selecting manually using DAPI channel exclusively (Appendix Fig. S3A,B). During all analysis, mice genotype and channel of interest were blinded. Analysis and quantification of images was performed using the QuPath software(Bankhead et al, 2017). Analysis workflows are available in Appendix Fig. S3. Briefly, the percentage of Collagen VI area per tissue section was quantified using pixel classifier/thresholder for the Col6 channel (Appendix Fig. S3C). The number of Lgr5 RNAscope spots per area was quantified using the RNAscope code detailed in Appendix Fig. S3D. The percentage of positive cells positive to different marker (E-cadherin, Ki67 or S100A9), was quantified using the STARDIST Qupath extension for cell detection (https://github.com/stardist/stardist) and specific classifiers were created for each marker (Appendix Fig. S3E). To determine Ly6a intensity, a coarse-grain analysis of Ly6a immu-nostaining was performed (Appendix Fig. S3F).

## Quantification of fecal and serum lipocalin-2 by ELISA

Frozen fecal samples collected at day 0, 4, 7, and 9 during DSS-induced colitis experiment were reconstituted in PBS containing 0.1% Tween-20 (100 mg/ml), vortexed at 4 °C for 20 min and centrifuged 10 min at 12,000 rpm at 4 °C. Clear supernatants were collected and stored at −20 °C until analysis. Fecal lipocaline-2 was quantified using the Duoset murine Lcn-2 ELISA kit (R&D Systems) accordingly to the manufacturer's instructions.

## qRT-PCR

Mice tissues were disrupted using a Lysing Matrix tube (MP Biomedicals) and homogenized in a tissue lyser (Qiagen) 2 × 2 min,

followed by RNA extraction using RNA easy mini extraction kit (Qiagen). RNA concentration was measured by spectrometry, and 1 mg of total RNA was used for reverse transcription. We then used a 1/10 dilution of the cDNA to perform quantitative real-time PCR using SYBR MasterMix (Life Technology) on a QuantStudio 6 Real-Time PCR System (Applied Biosystem). mRNA level in triplicate was normalized using Ribosomal protein S9 (RSP9) and Eukaryotic Translation Elongation Factor 1 Alpha 1 (EEF1A1) and Results were expressed as $2^{(-DDCt)}*100\%$. Primers used are listed in Table EV1.

## Western blot

Mice tissues were lysed 30 min at 4 °C on a rotating wheel in IP buffer (0.5% NP40; 500 mM Tris-HCl, pH=7.4; 20 mM EDTA; 2 mM benzamidine; 10 mM NaF and a cocktail of protease inhibitors). The suspended tissues were then put in a Lysing Matrix tube (MP Biomedicals) and homogenized in a tissue lyser (Qiagen) 2 × 2 min at 20 Hz. Homogenates were then pelleted at 5000 r.p.m. for 3 min at 4 °C. Supernatants were subjected to preclearing with G Sepharose beads, and protein amount was quantified by BCA assay (Pierce). For the immunoprecipitation assay, samples were incubated with G sepharose beads and goat anti-Antxr2 antibody overnight. After protein concentration normalization, samples were boiled in Laemmli buffer for 5 min before analysis by SDS–PAGE using 4–20% Bis-Tris gradient gels under reducing conditions and western blotting with rabbit anti-collagen VI antibody (dilution 1/1000), rat anti-Antxr2 (dilution 1/2), mouse anti-V5 antibody (dilution 1/2000), anti-Bcatenin (dilution 1/000) and mouse anti-Actin antibody (dilution 1/1000).

## Cells and plasmids

RPE1 (ATCC CRL-4000) were grown in Dulbecco's Modified Eagle Medium (Sigma Life Science) supplemented with 10% FBS, penicillin, and streptomycin (GIBCO). Mouse CMG2 (isoform 4, Uniprot identifier P58335-4) was cloned into a pcDNA3.5/V5-HIS-TOPO expression vector. Deletion of exon 3 in this plasmid was generated by In-Fusion (Takara 638948) mutagenesis. Plasmids were transfected into cells for 48 h, using Mirius according to the manufacturer's protocol. When described, cells were treated with proteasome inhibitor MG132 (dilution 1/1000) overnight before cell collection. All cell lines used in this study were authenticated and confirmed mycoplasma negative as tested on a trimestral basis using the MycoProbe Mycoplasma Detection Kit CUL001B.

## Quantification and statistical analysis

Unless otherwise stated, each data point corresponds to a single measurement, and each triangle corresponds to the mean per mouse. Statistical analysis was carried out using Prism software. Data representations and statistical details can be found in the description of the Figures. For ANOVA analysis, $P$ values were obtained by post hoc tests to compare every mean and pair of means (Tukey's and Sidak's). Sample sizes were determined based on prior experiments and power calculations to ensure adequate statistical power. Variation within groups is depicted as standard to the mean, and variances were comparable across groups included in statistical comparisons. Experiments were randomized, and inves-tigators were blinded to group allocation during data collection and

> **The paper explained**
>
> **Problem**
>
> Hyaline Fibromatosis Syndrome (HFS) is one of over 7000 known rare human diseases and is caused by mutations in the *Capillary Morphogenesis Gene 2* (*CMG2*). Depending on the nature of the mutation, symptoms range from debilitating to fatal, with the most severe cases leading to death in infancy due to recurrent intractable diarrhea. The cause of this intestinal dysfunction has remained unknown, and a role for CMG2 in gut physiology has not been documented. In this study, we investigated the function of CMG2 in the intestine using a knockout mouse model of HFS.
>
> **Results**
>
> Unexpectedly, CMG2 knockout (CMG2KO) mice showed no signs of intestinal dysfunction under normal conditions: their gut morphology, weight, and overall health appeared normal. However, the intestine's ability to regenerate after injury—an essential function of the gut—was severely impaired. Using a well-established model of chemically induced colitis (DSS treatment followed by withdrawal), we observed that while wild-type mice recovered fully, CMG2KO mice failed to regenerate their colonic epithelium. Further analysis revealed that the transition from fetal-like cells to Lgr5$^+$ intestinal stem cells—a Wnt-dependent step critical for regeneration—was disrupted in the absence of CMG2.
>
> **Impact**
>
> This study uncovers a previously unrecognized role for CMG2 in intestinal regeneration and provides a molecular explanation for the severe diarrheal symptoms seen in HFS patients. These insights raise the possibility that patients with HFS could benefit from treatments currently used in other regenerative or inflammatory gut diseases, such as inflammatory bowel disease or Crohn's disease.

analysis. All available samples that met predefined quality criteria were included, and exclusions were limited to technical artifacts.

## Data availability

All details regarding the code used for immunofluorescence and all data source files for this study are available on Zenodo (https://doi.org/10.5281/zenodo.15834887).

The source data of this paper are collected in the following database record: biostudies:S-SCDT-10_1038-S44321-025-00295-3.

## Peer review information

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

## Acknowledgements

We would like to thank the EPFL Facilities (BioImaging and Optics Core Facility, Center of PhenoGenomics and Histology Core Facility) and all the members of the F.G.v.d.G. lab for discussions and suggestions. This work was supported by the Swiss National Science Foundation (310030_214874), by the Gelù Foundation and by the Foundation Les Mûrons.

## Author contributions

**Lucie Bracq**: Conceptualization; Data curation; Formal analysis; Validation; Investigation; Visualization; Methodology; Writing—original draft; Writing—review and editing. **Audrey Chuat**: Conceptualization; Validation; Investigation; Methodology. **Béatrice Kunz**: Conceptualization; Validation; Investigation; Methodology. **Olivier Burri**: Software. **Romain Guiet**: Software. **Julien Duc**: Software; Formal analysis. **Nathalie Brandenberg**: Conceptualization; Writing—review and editing. **F Gisou van der Goot**: Conceptualization; Resources; Data curation; Supervision; Funding acquisition; Validation; Visualization; Methodology; Writing—original draft; Writing—review and editing.

Source data underlying figure panels in this paper may have individual authorship assigned. Where available, figure panel/source data authorship is listed in the following database record: biostudies:S-SCDT-10_1038-S44321-025-00295-3.

## Disclosure and competing interests statement

The authors declare no competing interests.

# Expanded View Figures

**Figure EV1.  Cmg2 KO mice have normal guts under basal conditions.**

(A) Body weight of Cmg2<sup>WT</sup> and Cmg2<sup>KO</sup> over a 30 weeks period. Results are mean ± SEM. Each dot represents the mean of at least $n = 8$ mice per genotype. (B) Colon tissues under basal conditions, from 8-weeks-old *Cmg2*<sup>WT</sup> and *Cmg2*<sup>KO</sup> mice were stained for Hematoxylin/Eosin, Sirius Red and Alcian Blue. Representative images of at least $n = 8$ mice per genotype. Scale bar, 20 μm. (C) Number of crypts per 800 μm of tissue screened, (D) crypt lengths were quantified and showed as superplot. Results are mean ± SEM. Each dot represents a single measurement and each triangle represent the mean per mouse. At least $n = 5$ mice per genotype were quantified. *P* values obtained by unpaired two-tailed *t* test. (E) Colon lysates were analyzed by SDS–PAGE using 4–12% Bis-Tris gradient gels under reducing condition and western blotted against all the collagen VI. Migration of the molecular weight markers (in kDa) are indicated on the left. (F) CollagenVI/loading control ratio were quantified and normalized to the mean of Cmg2<sup>WT</sup>. Results are mean ± SEM, and $n = 12$ mice per genotype were quantified. *P* values obtained by unpaired two-tailed *t* test. (G) Colon tissues were immunostained with anti-collagen VI and DAPI. Representative image of $n = 6$ mice per genotype. Scale bar, 20 μm. (H) % of Collagen VI area per tissue section was quantified on $n = 6$ mice per genotype. Results are mean ± SEM. Each dot represents a single measurement and each triangle represent the mean per mouse. *P* values obtained by unpaired two-tailed *t* test. q PCR analysis of colonic tissues for (I) Col6a1, and (J) ISC markers Lgr5, Ascl2, Axin2 and Cyclin D1 genes were performed on $n = 8$ mice per genotype. *P* values obtained by unpaired two-tailed *t* test in (I) and by two-way ANOVA Šídák's multiple comparisons test in (J). (K) Colonic tissues were stained for RNAscope in situ hybridization against Lgr5 and (L) number of Lgr5 mRNA spots per μm² were quantified. (M) Colonic sections were stained anti-ki67, anti-E-cadherin and DAPI. (N) % of E-cadherin+ epithelial cells, and (O) % of E-cadherin + /Ki67+ epithelial cells, were quantified on $n = 8$ mice per genotype. *P* values obtained by unpaired two-tailed *t* test. (P) qPCR analysis of colonic tissues for inflammatory cytokines genes was performed on $n = 8$ mice per genotype. *P* values obtained by two-way ANOVA Šídák's multiple comparisons test.

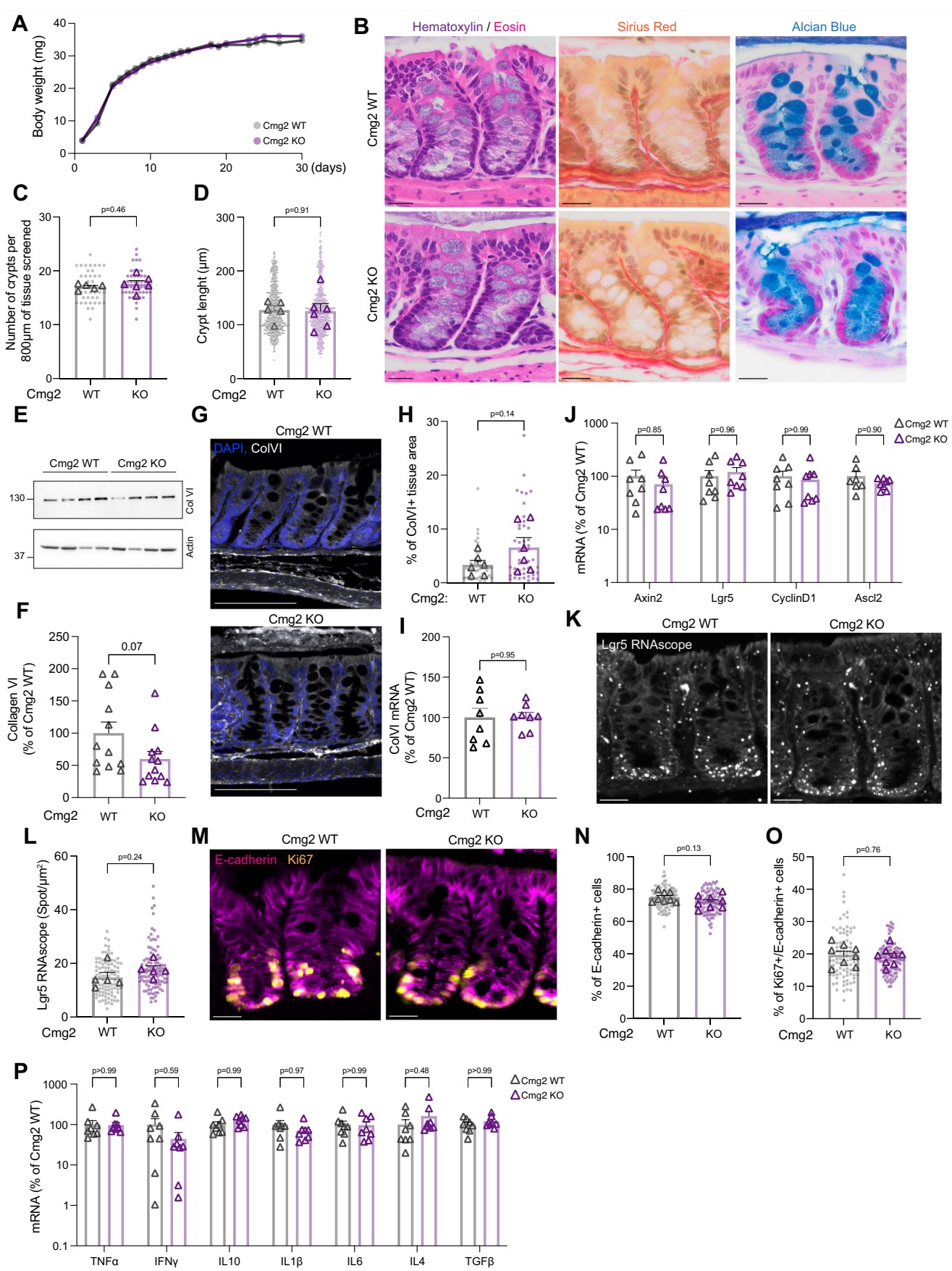

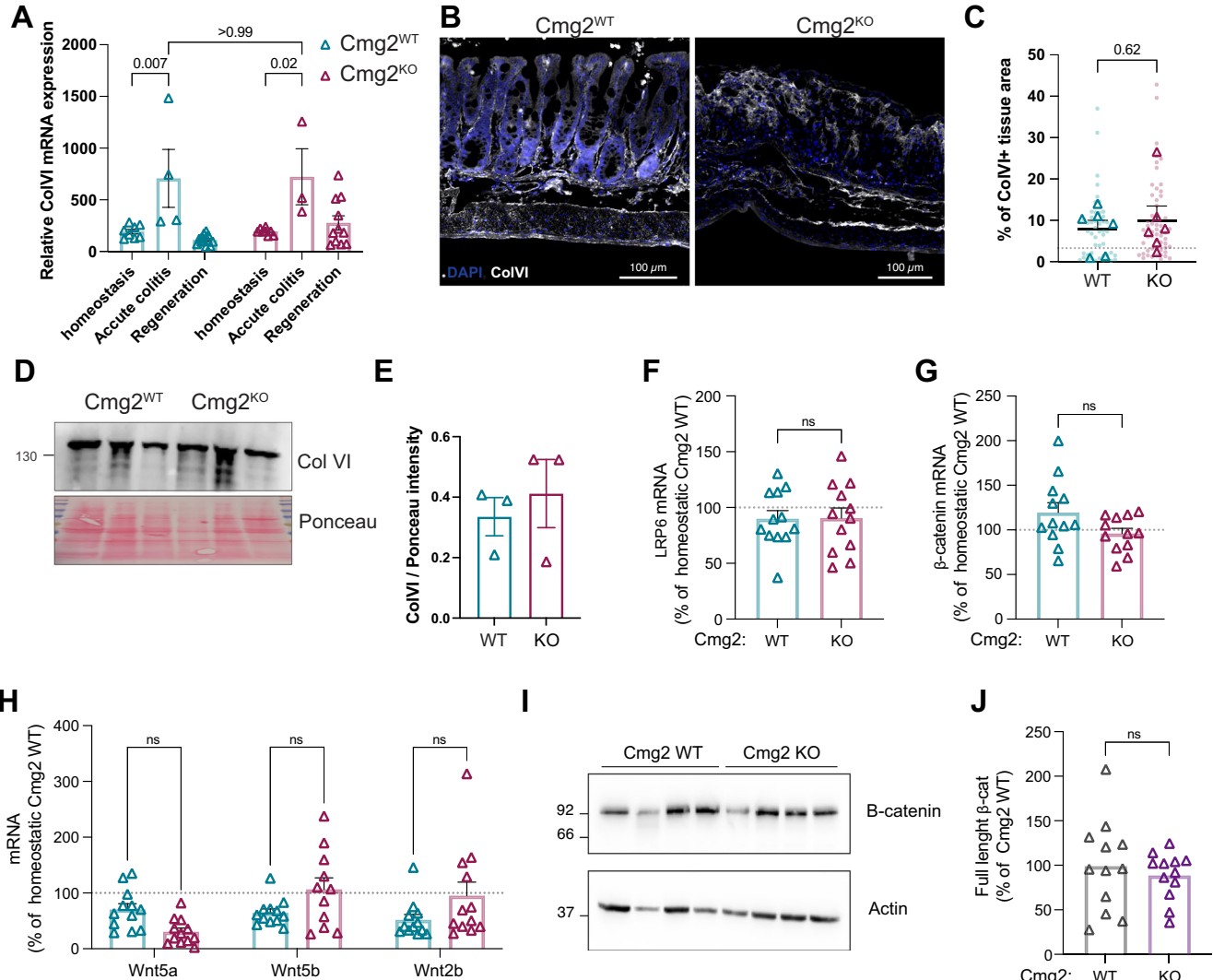

**Figure EV2. Collagen and Wnt players in Cmg2^KO mice.**

(A) qPCR analysis of colonic tissues from *Cmg2*^WT and *Cmg2*^KO during homeostasis, after 7days of DSS or 3 days after DSS withdrawal for Col6a1. Results are mean ± SEM. Each symbol represents the mean per mouse. At least n = 4 mice per genotype were quantified. *P* values obtained by two-way ANOVA Šídák's multiple comparisons test. (B) Colon tissues from Cmg2^WT and Cmg2^KO mice 3 days after DSS withdrawal (day 10) were immunostained with anti-collagen VI and DAPI. Representative image of n = 6 mice per genotype. Scale bar, 100 μm. (C) % of Collagen VI area per tissue section was quantified on n = 6 mice per genotype. Results are mean ± SEM. Each dot represents a single measurement and each triangle represent the mean per mouse. *P* values obtained by unpaired two-tailed *t* test. (D) colonic lysates from Cmg2^WT and Cmg2^KO mice 3 days after DSS withdrawal (day 10) were analyzed by SDS–PAGE using 4–12% Bis-Tris gradient gels under reducing condition and western blotted against all the collagen VI. Migration of the molecular weight markers (in kDa) are indicated on the left. (E) CollagenVI/Ponceau ratio were quantified. Results are mean ± SEM, and n = 3 mice per genotype were quantified. (F–H) qPCR analysis of colonic tissues from Cmg2^WT and Cmg2^KO mice 3 days after DSS withdrawal (day 10) for (F) LRP6, (G) β-catenin and (H) Wnt ligand, Results are mean ± SEM. Each symbol represents the mean per mouse. The dotted line represents the mean of homeostatic Cmg2^WT mice. n = 12 mice per genotype was quantified. *P* values obtained by two-tailed unpaired *t* test in (F, G) or by two-way ANOVA Šídák's multiple comparisons test in (H). (I) Colon lysates were western blotted against β-catenin. (J) β-catenin/loading control ratio were quantified and normalized to the mean of Cmg2^WT. Results are mean ± SEM, and n = 12 mice per genotype were quantified. Each triangle represents the mean per mouse. *P* values obtained by unpaired two-tailed *t* test.

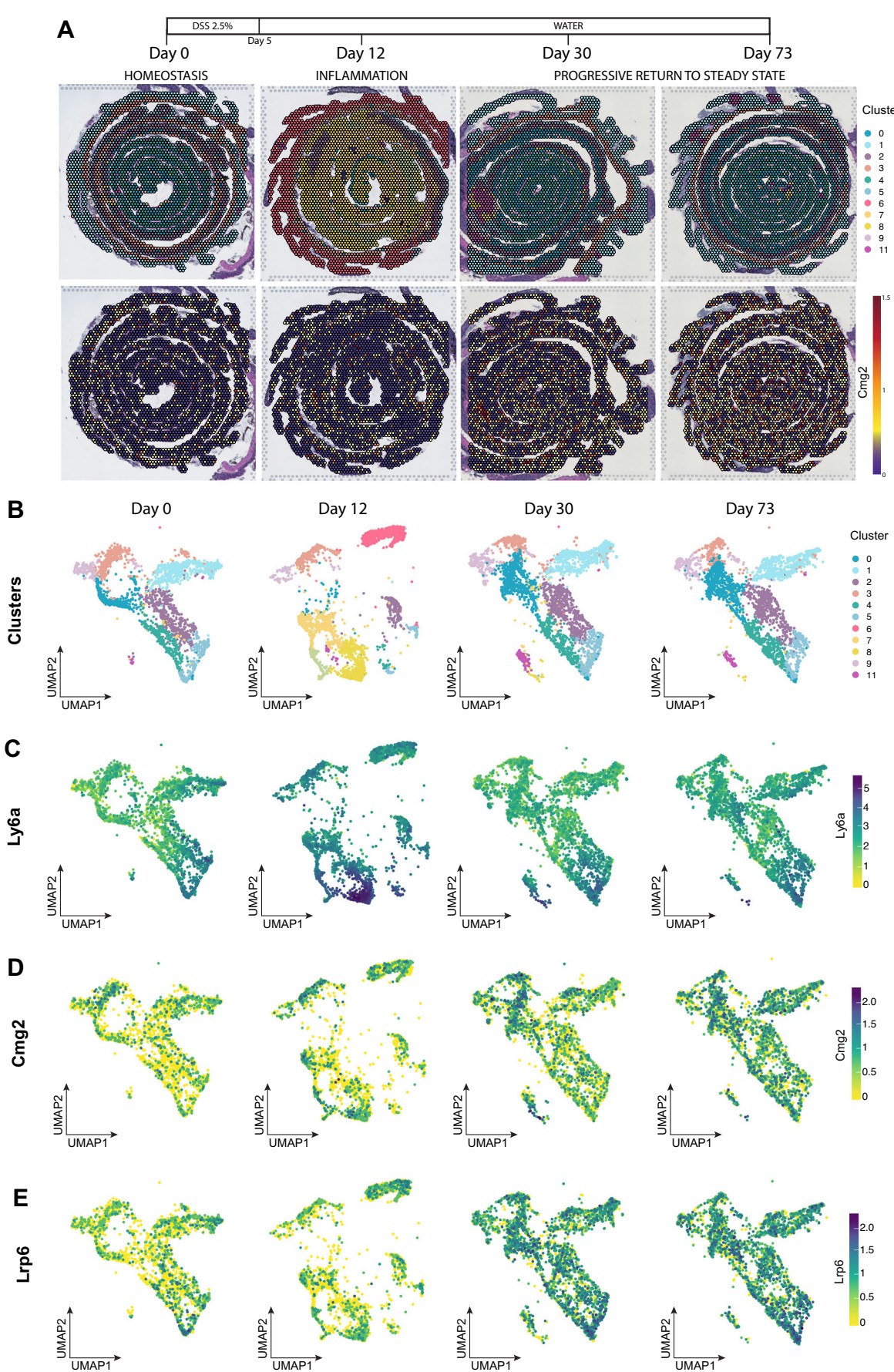

◀ **Figure EV3. Spatial transcriptomics of DSS-treated mice throughout the post-treatment recovery course for up to 73 days.**

(A) Clusters of VISIUM spots (upper panel) or Cmg2 expression (lower panel) coupled with tissue H&E staining, replotted from raw data published in Mayassi et al, 2024 (Data ref: Mayassi et al, 2024a). (B) UMAP clustering of VISIUM data. (C–E) Ly6a (C), Cmg2 (D) and Lrp6 (E) expression on intestinal tissue from VISIUM data.

