## [Peer Review File · EMBO Molecular Medicine]

Injury-Induced Intestinal Stem Cell Renewal Requires Capillary Morphogenesis Gene 2

Lucie Bracq, Audrey Chuat, Béatrice Kunz, Olivier Burri, Romain Guiet, Julien Duc, Nathalie Brandenburg and F. Gisou van der Goot

Corresponding author: F. Gisou van der Goot (gisou.vandergoot@epfl.ch)

Review Timeline:

Transferred from Review Commons:	27th May 25
Editorial Decision:	23rd Jun 25
Revision Received:	14th Jul 25
Accepted:	22nd Jul 25

Editor: Lise Roth

Transaction Report:

This manuscript was transferred to EMBO Molecular Medicine following peer review at Review Commons.

Review #1**1. Evidence, reproducibility and clarity:****Evidence, reproducibility and clarity (Required)**

In this work, Bracq and colleagues provide clear evidence that the persistent diarrhoea seen in a mouse model of Hyaline Fibromatosis Syndrome is related to the inability of their intestinal epithelium to properly regenerate. This is very clear and of immediate impact. This aspect of the paper, I think, is ready for publication, and would merit immediate dissemination on its own. It is great that the manuscript is in bioRxiv already.

I am not so thoroughly convinced about the mechanism that the author propose to explain the incapacitation of Cmg2[KO] intestinal stem cells to function properly. The authors propose that it is due to their inability to transduce Wnt signals, and while this is plausible, I think there are few things that the paper should contain before this can be proposed firmly:

Point #1

The mouse mutant is just described as 'KO', referring to the previous work by the authors. The cited work simply states that this is a zygotic deletion of exon 3, which somehow leads to a decrease in protein abundance that is almost total in the lung but not so clear in the uterus. Exon 3 happens to be 72 bp long [https://www.ncbi.nlm.nih.gov/nucore/NM_133738], so its deletion (assuming there are no cryptic splicing sites used) leads to an internal in-frame deletion of 24 amino acids. So, at best, this 'KO' is not a null, but a hypomorphic allele of context-dependent strength. Unfortunately, neither the previous work nor this paper (unless I have missed it!) contains information provided about the expression levels of Cmg2 in the intestine of KO mice - nor which cell types usually express it (see below). I think that using anti Cmg2 in WB and immunohistofluorescence of with ISC markers with intestine homogenate/sections of wild-type and mutant mice would be necessary to set the stage for the rest of the work.

Point #2

Connected to the previous point, the expression pattern of Cmg2 in the intestine is not described. Maybe this is already established in the literature, but the authors do not refer to the data. This is important when considering that the previous work of the authors suggests that Cmg2 might contribute to Wnt signalling transduction through physical, cis interactions with the Wnt co-receptor LRP6. Therefore, one would expect that Cmg2 would

be cell-autonomously required in the intestinal stem cells.

Point #3

The authors establish that the regenerating crypts of Cmg2[KO] mice are unable to transduce Wnt signalling, but it is not clear whether this situation is provoked by the DSS-induced injury or existed all along. Can Cmg2[KO] intestinal stem cells transduce Wnt signalling before the DSS challenge? If they were, it might suggest that the 'context-dependence' of the Cmg2 role in Wnt signalling is contextual not only because of the tissue, but because of the history of the tissue or its present structure. It would also suggest that Cmg2 mutant mice, unless reared in a germ-free facility for life, would eventually lose intestinal homeostasis, and maybe suggest the level of intervention/monitoring that HFS patients would require. It might also provide an explanation in case Cmg2 was not expressed in ISCs - if the state of the tissue was as important as the presence of the protein, then the effect on Wnt transduction could be indirect and therefore it might not be required cell-autonomously.

I think points 1 and 2 are absolutely fundamental in a reverse genetics investigation. Point 3 would be nice to know but the outcome would not change the tenet of the paper. I believe that the work needed to deal these points can be performed on archival material. I do not think the mechanism proposed can be taken from 'plausible' to 'proven' without proposing substantial additional investigation, so I will not suggest any of it, as it could well be another paper.

A few minor points picked along the way:

1. Figure 1 legend says "In (c), results are mean {plus minus} SEM" - this seems applicable to (d) as (c) does not show error whiskers.
2. Figure 1 legend says "(d) Body weight loss, (f) the aspect of the feces and presence of occult blood were monitored and used for the (e) DAI. Results are mean {plus minus} SEM. Each dot represents the mean of n = 12 mice per genotype". This part looks like has suffered some rearrangement of words. The first instance of (f) should be (e), I guess, and I am not sure what "(e) DAI" means. And for (e), "mean {plus minus} SEM" does not seem applicable. This needs some light revision.
3. Figure 1H legend does not say which statistical test was made in the survival experiment in (h) - presumably log-rank? A further comment on the survival statistics: euthanised animals should not be counted towards true mortality when that is what is recorded as an 'event'. They should be right-censored. However, in this case, reaching the euthanasia

criterion is just as good an indicator of health as mortality itself. So, simply by changing the Y axis from 'survival' to 'event-free survival' (or something to that effect), where 'events' are either death or reaching the euthanasia criterion, leaves the analysis as it is, and authors do not need to clarify that figure 1H shows "apparent mortality", as it is straightforward "complication-free survival" (just not entirely orthogonal to weight loss).

4. Some density measurements are made unnecessarily on arbitrary units (per field of view) - this should be simple to report in absolute measures (i.e. area of tissue screened or, better still, length of epithelium screened).

5. Figure 2E should read "percent involvement"

6. Figure 2J should read "lipocalin..."

7. In section "CMG2 Is Dispensable for YAP/TAZ-Mediated Reprogramming to Fetal-Like Stem Cells", the authors write ""We measured the mRNA levels of two additional YAP target genes, Cyr61 and CTGF...". I presume the "additional" is because Ly6a is also a target of YAP/TAZ, but if the reader does not know, it is puzzling. I would suggest to make this link explicit.

8. In Figures S2, 3 and S3, I think that the measures expressed as "% of homeostatic X in WT" really mean "% of average homeostatic X in WT". This should be made clear somewhere.

9. In panel C, the nature of the data is not entirely clear. First, the corresponding part of the legend says "Representative images of n=4 mice per genotype" which I presume should refer to panel B. Then, the graph plots 4 data points, which suggests that they correspond to 4 mice - but how many fields of view? Also, the violin plot outline is not described - I presume it captures all the data points from the coarse-grained pixel analysis, but it should be clarified.

10. In Figure 3H and 3I, I would suggest to add the 7+3 timepoint where the data come from.

11. In section "CMG2 Is Critical for Restoring the Lgr5+ Intestinal Stem Cell Pool", the authors say "...The mRNA levels of ... LRP6, β -catenin (Fig. S3a-b), and Wnt ligands (Wnt5a, 5b, and 2b) were comparable between the colons of Cmg2WT and Cmg2KO mice (Fig. S3c)..." without clarifying in which context - one needs to read the figure legend to realise this is "timepoint 7+3". I suggest to add "in the recovery phase" or "in regenerating colons" or something shorter, just to guide the reader.

12. Like with the previous point, it is not clear when the immunohistofluorescence of β -catenin is made - not even in the legend, as far as I could see. The only hint is that authors say "the nuclei of cells in the atrophic crypts of Cmg2KO..." with 'atrophic' probably indicating again the 7+3 timepoint.

13. A typo in the discussion: tunning for tuning.

14. In the discussion, the authors talk about the 'CMG2' protein (all caps - formatting

convention for human proteins) but before they were referring to 'Cmg2' (formatting convention for mouse proteins). That is fine but some of the statements where "CMG2" is used clearly refer to observations made in the mouse.

15. Typos in methods: "antigen retrieval by treating [with] Proteinase K"; "Image acquisition and analyze [analysis]"; "All details regarding code[s] used for immunofluorescence analysis"

****Referees cross-commenting****

*this session contains comments from ALL the reviewers"

Rev2

Points 1 and 2 made by Referee 1 (and point 4 of Referee 3) appear most reasonable, and if not already done should be.

I also noted the more severe morphology of DSS damaged epithelium shown in Fig 2a noted by Referee 3 - and this I agree is a confounding factor. But overall, multiple lines of evidence were assembled to show that the KO mice and WT mice suffered DSS-induced colitis with equal severity - and with closely equal severity of damage to the intestinal epithelium (though the image in Fig 2a is disturbing). For my part, the concern is understandable but likely not operating in a confounding way. And the evidence for the reprogramming of the damaged epithelium into "fetal-like stem cells" (the 1st step in restitution of lost stem cells) occurs in both WT and KO mice - and these data are strong. For this reader, the block convincingly shows up for KO mouse at the WNT dependent step

Rev 3

This reviewer remains sceptical. I agree the authors performed the experiment well to confirm that DSS dosing was as equivalent as possible across the study. But DSS acts to induce colitis because it is concentrated in the colonic lumen as water is absorbed. Also ECM responses and remodelling are a central part of colitis models. And my concern is that the actual exposure in the KO group is influenced by transit of faeces/DSS is secondary to the known action of CMG2 on collagen deposition. The consequence of this being a protracted damage phase in which a restoration of adult stem cells would not be expected and leading to epithelial failure.

However, we differ. I might propose that the authors are asked to investigate and confirm expression of CMG2 in the epithelium and to repeat the analysis of collagen levels they

performed on untreated CMG2 KO mice on colons from CMG2 KO mice having received DSS to see if these differ from controls.

Rev 1

Both reviewer #2 and reviewer #3 make relevant points, from the point of view of extracting as much biological knowledge as we can from the observations reported in the manuscript.

Reviewer #2 suggestion to use Cmg2[KO] organoids to investigate the dependence of Wnt transduction on Cmg2 is the type of experiments I refrained to propose. However, I think the "skeleton" of the mechanism is there and is reasonably solid. Fleshing it out may well be another paper.

I agree with Reviewer #3 objections to the timing and severity of the DSS damage. However, I am not sure how much they invalidate the main tenet of the paper:

- DSS may affect Cmg2[KO] more severely, but the overall disease score is comparable during the DSS treatment. If this severity was enough to be the main driver of the phenotype, it should have left a mark in the Histological and Disease activity scores. In this regard, I think it would be helpful if the authors provided an expanded version of Figure 2A with examples of the different levels of "Crypt damage" scored, and the proportions for each. This could be in the supplementary material and would balance the impressions induced by a single image.

- If DSS affected the recovery, this would also be compatible with having a more severe histological phenotype (which is not shown overall, just in Fig 2A) because one would also expect the tissue to attempt regeneration during the 7 days of DSS treatment.

- The only objection that I find difficult to argue is the effective duration of the treatment. If indeed peristalsis is affected, it may be that during the 'recovery' phase there is still DSS in the intestine. This could be perhaps verified using a DS detection assay (e.g. <https://arxiv.org/pdf/1703.08663>) on the intestinal contents or the faeces of the mice during the 3-day recovery period.

I think of what the aim of scholarly publication is, with this paper, and I find myself going back to a statement of the authors' discussion - that this work suggests that infants risking death may be offered (compassionate, I guess) IBD treatment. What does this hinge upon? I think, on the basic observation that diarrhoea (in the mouse model) is not intrinsic but caused by an inflammation-promoting insult. Is this substantiated? I think it is. Could we

learn more biology from this disease model, about Wnt and about how ECM affects tissue regeneration? Certainly. Can this learning wait? I believe it can.

2. Significance:

Significance (Required)

In this work, Bracq and colleagues provide clear evidence that the persistent diarrhoea seen in a mouse model of Hyaline Fibromatosis Syndrome is related to the inability of their intestinal epithelium to properly regenerate. This is very clear and of immediate impact. For instance, the authors themselves point at the possibility of applying treatments for Inflammatory Bowel Disease to HFS patients. While what happens in a mouse model is not necessarily the same as in human patients, the fact that persistent diarrhoea is a life-threatening symptom in HFS make this proposal, at least in compassionate use of the therapies and until its efficacy is disproven, very plausible. This is a clear gap of knowledge that addresses an unmet medical need.

I find that the work shows clearly that HFS mouse model subjects have normal intestinal function until challenged with a standard chemically-induced colitis. Then, the histological and health deterioration of the HFS mouse model is clear in comparison with normal mice, which can regenerate appropriately. This is shown with a multiplicity of orthogonal techniques spanning molecular, histological and organismal, which are standard and very well reported in the paper.

The authors propose a specific cellular and molecular mechanism to explain the incapacity of the intestinal epithelium in the mouse model of HFS to regenerate. According to this mechanism, the protein Cmg2, whose mutation causes HFS in humans, would be necessary for intestinal stem cells to transduce the signal of Wnt ligands and therefore support their behaviour as regenerative cells. This mechanism is plausible, but more basic and advanced work would be needed to take it as proven.

This work would be of interest to both the clinical, biomedical, and basic research communities interested in rare diseases, the gastrointestinal system, collagen and extracellular matrix, and Wnt signalling.

My general expertise is in developmental and stem cell biology using reverse genetics, transgenesis and immunohistological and molecular methods of data production, and lineage tracing, digital imaging and bioinformatic analytical methods; I work with *Drosophila melanogaster* and its adult gastrointestinal system.

3. How much time do you estimate the authors will need to complete the suggested revisions:

Estimated time to Complete Revisions (Required)

(Decision Recommendation)

Between 1 and 3 months

4. Review Commons values the work of reviewers and encourages them to get credit for their work. Select 'Yes' below to register your reviewing activity at Web of Science Reviewer Recognition Service (formerly Publons); note that the content of your review will not be visible on Web of Science.

Yes

Review #2

1. Evidence, reproducibility and clarity:

Evidence, reproducibility and clarity (Required)

The paper uses mice lacking Capillary Morphogenesis Gene 2 (CMG2- KO) mice to investigate the pathogenic mechanism underlying the protein losing enteropathy seen in children with severe Hyaline Fibromatosis Syndrome. Significance of the work is further enhanced as the intestinal phenotype induced by CMG2-KO provided a model system (with robust validated tools) for testing newly emerging (and paradigm shifting) ideas in mechanisms of tissue regeneration after injury - generalizable to tissue restitution beyond the intestine.

The study shows that in the mouse colon CMG2 plays a critical role in recovery from mucosal/epithelial damage chemically induced by dextran-sulfate-sodium (DSS). Mice lacking CMG2 failed to recover from DSS colitis with no evidence for restitution of the DSS-damaged epithelium. WT mice recovered after DSS removal.

The first step in restitution of epithelial damage in the intestine, when the epithelial stem-cell populations are depleted as in this model of DSS colitis, occurs by the transformation of surviving differentiating/differentiated epithelial cells back into a stem-cell-like (fetal-cell-like) state. This step in the process was found to occur normally in the CMG2 KO mouse. The block in restitution was located to the step where de-differentiated (fetal-cell-

like) colonocytes are induced back into their WNT-dependent proliferative state - thus replenishing the normally proliferating stem (LGR5+) cells of the colonic crypt. The reason for this failure is explained by a defect in WNT signaling in the injured colons of CMG2 KO mice, as assessed by failure of β -catenin translocation into the nucleus of barrier epithelial cells - a down-stream effect of WNT signaling and consistent with the dependence on CMG2 for WNT signaling in other experimental systems.

The study is overall well designed, meticulously carried out, and with clear and convincing results that are most reasonably and thoughtfully interpreted. The paper makes a meaningful contribution to the field. It models an experiment of nature to test, delineate, and verify disease pathogenesis and a newly revised mechanism for mucosal tissue repair.

For this reader, one additional thought comes to mind. If I understand the field correctly it would be informative to know with greater confidence where - in what cell type, epithelial or mesenchymal - the CMG2-LRP6-WNT interaction occurs.

After injury the CMG2-KO mouse epithelium exhibits defective WNT signal transduction - as evidenced by failure of β -catenin to translocate into the nucleus. At first glance, this result is a disconnect with the paper by van Rijin that claims the defect in Hyaline Fibromatosis Syndrome cannot be due to loss of CMG2 expression/function in the barrier epithelial cell - a claim based on the mostly normal phenotypes of human CMG2 KO duodenal organoids. But the human organoids studied in the van Rijin paper, like all others, are established and cultured in very high WNT conditions, perhaps obscuring the lack of the CMG2-LRP6-WNT interaction. And in fact, the phenotypes of these human CMG2-KO duodenoids were not entirely normal - the CMG2-KO stem-like organoids (even when cultured in high WNT/R-spondin conditions) developed abnormal intercellular blisters consistent with a defect in epithelial structure/function - of unknown cause and not investigated.

I think it would be informative to prepare colon organoids (and duodenoids) from WT and CMG2-KO mice to quantify their WNT dependency during establishment and maintenance of the stem-like (and WNT-dependent) state. If CMG2 acts within the epithelial cell to affect WNT signaling (regardless of WNT source), organoids prepared from colons of CMG2-KO mice would require more WNT in culture media to establish and maintain the stem cell proliferative state - when compared to organoids prepared from WT mice. This can be quantified (and confirmed molecularly by transgene expression if successful). Enhanced dependency of high concentrations of exogenous WNT would be evidence for a primary defect in WNT-(LRP2)-CMG2 signal transduction localized to the epithelial barrier cell -

thus addressing the apparent discrepancy with the van Rijn paper - and for my part, advancing the field. And the discovery of a defect in the epithelium itself for WNT signal transduction would implicate a biologically most plausible mechanism for development of protein losing enteropathy.

By no means do I consider these experiments to be required for publication (especially if considered to be incremental or already defined - WNT-CMG2 is not my field of research). This study already makes a meaningful contribution to the field as I state above.

But in the absence of new experimentation, the issue should probably be discussed in greater depth.

2. Significance:

Significance (Required)

The study is overall well designed, meticulously carried out, and with clear and convincing results that are most reasonably and thoughtfully interpreted. The paper makes a meaningful contribution to the field. It models an experiment of nature to test, delineate, and verify disease pathogenesis and a newly revised mechanism for mucosal tissue repair.

3. How much time do you estimate the authors will need to complete the suggested revisions:

Estimated time to Complete Revisions (Required)

(Decision Recommendation)

Between 1 and 3 months

4. Review Commons values the work of reviewers and encourages them to get credit for their work. Select 'Yes' below to register your reviewing activity at Web of Science Reviewer Recognition Service (formerly Publons); note that the content of your review will not be visible on Web of Science.

Yes

Review #3 -

1. Evidence, reproducibility and clarity:

Evidence, reproducibility and clarity (Required)

This manuscript has a good rationale in trying to understand why infants with an inherited condition, Hyaline Fibromatosis Syndrome, that is primarily associated with turnover and deposition of extracellular collagen also develop severe diarrhoea that can contribute to their premature death. The premise is that the causative germline mutated gene, CMG2/ANTRX2, may have a functional role in colonic epithelium in addition to controlling the ECM composition. There is little background information but one study has shown no primary defect in epithelial organoids grown from patients with the syndrome. This leads the authors to wonder if non-homeostatic, conditions might reveal a function role for the gene in regeneration.

The authors' approach to test the hypothesis is to use a mouse germline knockout model and to induce colitis and regeneration by the established protocol of introducing dextran sodium sulfate (DSS) into the drinking water for five days. In brief there is no phenotype apparent in the untreated knockout (KO) but these animals show a more severe response to DSS that requires them to be killed by 10 days after the start of treatment. This effect following phenotypic characterisation of the colonic epithelium is interpreted as showing the CMG2 is a Wnt modifier required for the restoration of the intestinal stem cell population in the final stages of repair.

The experiment and analysis seem reasonably well executed - although a few specific comments follow below. The narrative is simple and easy to understand. However, there are significant caveats that cast doubts on the interpretation made that loss of CMG2 impairs the transition of colonic epithelial cells from a fetal like state to adult ISCs.

2. Significance:

Significance (Required)

1. First there is only a single approach and single type of experiment performed. There is a lack of independent validation of the phenotype and how it is mediated.
2. The DSS dose in this kind of experiment is often determined empirically in individual units. Here the 3% used is within published range but at upper end. The control animals show a typical response with symptoms of colitis worsening for 2-3 days after the removal of DSS and then recovery commonly over another 5-7 days.

Here the CMG2 KO mice fail to recover and are killed by 9 or 10 days. The authors attempt to exploit the time course by identifying normal initial (7days) and defective late (10days) repair phases in KO animals when compared to controls. It is from this comparison that

conclusions are drawn.

However, the alternative interpretation might be that the epithelium of KO animals is so badly damaged, and indeed non-existent (from viewing Fig2a), that it is incapable of mounting any other response other than death and that the profiling shown is of an epithelium in extremis. The repair capability and dynamics of the KO would have been better tested under more moderate DSS challenge, if this experiment had been regarded as a pilot rather than as definitive.

3. The animals used were young (8 weeks) and lacked any obvious defect in collagen deposition. Does this change with treatment? Even if not, is it possible that there is a defect in peristalsis or transit time of gut contents, resulting in longer dwell times and higher effective dose of DSS to the KO epithelium?

4. Is CMG2 RNA and protein expressed in the colonic epithelium? It is not indicated or tested in the submitted manuscript. This reviewer struggled to find evidence, notably it did not seem to be referenced in the organoid paper they reference in introduction (ref 13).

****Specific comments:****

Figure 3 c-e and associated text are confusing. In c the Y scale seems inappropriate to show percentages up to 15,000%.

In d and e the use of percentages may be correct. However, it is claimed in text that Cty61 and CTFG are upregulated in the KO. That is not what the plots appear to show as they compare to WT untreated cells, in which case the KO have not downregulated these genes in the way the controls have.

3. How much time do you estimate the authors will need to complete the suggested revisions:

Estimated time to Complete Revisions (Required)

(Decision Recommendation)

Cannot tell / Not applicable

4. Review Commons values the work of reviewers and encourages them to get credit for their work. Select 'Yes' below to register your reviewing activity at Web of Science Reviewer Recognition Service (formerly Publons); note that the content of your review will not be visible on Web of Science.

No

We thank the reviewer for their constructive comments and the fair and interesting discussion between reviewers.

Reviewer #1

We are delighted to read that the reviewer finds the manuscript “*very clear and of immediate impact [...] and ready for publication*” regarding this aspect. We have toned down the conclusion, proposing rather than concluding that “*the incapacitation of *Cmg2*[KO] intestinal stem cells to function properly [...] is due to their inability to transduce Wnt signals*”.

We have addressed the 3 points that were raised as well as the minor comments.

Point #1

The mouse mutant is just described as 'KO', referring to the previous work by the authors. The cited work simply states that this is a zygotic deletion of exon 3, which somehow leads to a decrease in protein abundance that is almost total in the lung but not so clear in the uterus. Exon 3 happens to be 72 bp long [https://www.ncbi.nlm.nih.gov/nucore/NM_133738], so its deletion (assuming there are no cryptic splicing sites used) leads to an internal in-frame deletion of 24 amino acids. So, at best, this 'KO' is not a null, but a hypomorphic allele of context-dependent strength.

Unfortunately, neither the previous work nor this paper (unless I have missed it!) contains information provided about the expression levels of *Cmg2* in the intestine of KO mice - nor which cell types usually express it (see below). I think that using anti *Cmg2* in WB and immunohistofluorescence of with ISC markers with intestine homogenate/sections of wild-type and mutant mice would be necessary to set the stage for the rest of the work.

We now provide an explanation and characterization of the *Cmg2*^{KO} mice. Exon 3 indeed only encodes a short 24 amino acid sequence. This exon however encodes a β -strand that is central to the vWA domain of CMG2, and therefore critical for the folding of this domain. As now shown in Fig. S1c, *CMG2* Δ exon3 is produced in cells but cleared by the ER associated degradation pathway, therefore it is only detectable in cells treated with the proteasome inhibitor MG132, at a slightly lower molecular weight than the full-length protein. This is consistent, and was inspired by the fact that multiple Hyaline Fibromatosis missense mutations that map to the vWA domain lead to defective folding of CMG2, further illustrating that this domain is very vulnerable to modifications. In Fig. S1c, we moreover now show immunoprecipitation of *Cmg2* from colonic tissue of wild-type (WT) and knockout (KO) mice, which confirm the absence of *Cmg2* protein in *Cmg2*^{KO} samples.

Point #2

Connected to the previous point, the expression pattern of *Cmg2* in the intestine is not described. Maybe this is already established in the literature, but the authors do not refer to the data. This is important when considering that the previous work of the authors suggests that *Cmg2* might contribute to Wnt signalling transduction through physical, cis interactions with the Wnt co-receptor LRP6. Therefore, one would expect that *Cmg2* would be cell-autonomously required in the intestinal stem cells.

The expression pattern of *Cmg2* in the gut has not been characterized and is indeed essential to understanding its function. To address this gap, we now added a figure (Fig. 1) providing data from publicly available RNA-seq datasets and from our RNAscope experiments on *Cmg2*^{WT} mice. Of note, we unfortunately have never managed to detect *Cmg2* protein

expression by immunohistochemistry of mouse tissue with any of the antibodies available, commercial or generated in the lab.

In the RESULTS section we now mention:

To investigate Cmg2 expression in the gut, we first analyzed publicly available spatial and scRNA-seq datasets to identify which cell types express Cmg2 across different gut regions. Spatial transcriptomic data from the mouse small intestine and colon revealed that Cmg2 is broadly expressed throughout the gut, including in the muscular, crypt, and epithelial layers (Fig. 1A–C). To validate these findings, we performed RNAscope in situ hybridization targeting Cmg2 in the duodenum and colon of wild-type mice. The expression pattern observed was consistent with the spatial transcriptomics data (Fig. 1D–E). We then analyzed scRNA-seq data from the same dataset to assess cell-type-specific expression in the mouse colon. Cmg2 was detected at varying levels across multiple cell types, including enterocytes and intestinal stem cells, as well as mesenchymal cells, notably fibroblasts.

Of note for the reviewer, not mentioned in the manuscript, this wide-spread distribution of Cmg2 across the different cell types is not true for all organs. We have recently investigated the expression of Cmg2 in muscle and found that it is almost exclusively expressed in fibroblasts (so-called fibro-adipocyte progenitors) and very little in any other muscle cells, in particular fibers.

Interestingly also, as now mentioned in the manuscript and shown in Fig. S1, the ANTXR1 protein, which is highly homologous to Cmg2 at the protein level and share its function of anthrax toxin receptor, displayed a much more restricted expression pattern, being confined primarily to fibroblasts and mural cells, and notably absent from epithelial cells. This differential expression highlights a potentially unique and epithelial-specific role for Cmg2 in maintaining intestinal homeostasis.

Point #3

The authors establish that the regenerating crypts of Cmg2[KO] mice are unable to transduce Wnt signalling, but it is not clear whether this situation is provoked by the DSS-induced injury or existed all along. Can Cmg2[KO] intestinal stem cells transduce Wnt signalling before the DSS challenge? If they were, it might suggest that the 'context-dependence' of the Cmg2 role in Wnt signalling is contextual not only because of the tissue, but because of the history of the tissue or its present structure. It would also suggest that Cmg2 mutant mice, unless reared in a germ-free facility for life, would eventually lose intestinal homeostasis, and maybe suggest the level of intervention/monitoring that HFS patients would require. It might also provide an explanation in case Cmg2 was not expressed in ISCs - if the state of the tissue was as important as the presence of the protein, then the effect on Wnt transduction could be indirect and therefore it might not be required cell-autonomously.

We agree that understanding whether Cmg2^{KO} intestinal stem cells are intrinsically unable to transduce Wnt signals, or whether this defect is contextually induced following injury (such as DSS treatment), is a critical point.

As a first line of evidence, we show that under homeostatic condition, Wnt signaling appears largely intact in Cmg2^{KO} crypts, with comparable levels of β -catenin and expression levels of canonical Wnt target genes (e.g., *Axin2*, *Lgr5*) to those observed in WT animals (Figs. S1j-l and S3d-e). This indicates that Cmg2 is not essential for basal Wnt signaling under steady-state conditions.

These findings thus support the idea that the requirement for Cmg2 in Wnt signal transduction is context-dependent—not only at the tissue level but also temporally, being specifically required during regenerative processes or in altered microenvironments such as during inflammation or epithelial damage. This context-dependence may reflect changes in the

composition or accessibility of Wnt ligands, receptors, or matrix components during repair, where Cmg2 could play a scaffolding or stabilizing role.

These aspects are now discussed in the text.

I think points 1 and 2 are absolutely fundamental in a reverse genetics investigation. Point 3 would be nice to know but the outcome would not change the tenet of the paper. I believe that the work needed to deal these points can be performed on archival material. I do not think the mechanism proposed can be taken from 'plausible' to 'proven' without proposing substantial additional investigation, so I will not suggest any of it, as it could well be another paper.

We have addressed points 1 and 2, and provided evidence and discussion for Point 3.

Minor points

1- Figure 1 legend says "In (c), results are mean {plus minus} SEM" - this seems applicable to (d) as (c) does not show error whiskers.

We thank the reviewer for picking up this error. We modified : "In (c), results are median" and "In (d, f and g) Results are mean \pm SEM."

2- Figure 1 legend says "(d) Body weight loss, (f) the aspect of the feces and presence of occult blood were monitored and used for the (e) DAI. Results are mean {plus minus} SEM. Each dot represents the mean of n = 12 mice per genotype". This part looks like has suffered some rearrangement of words. The first instance of (f) should be (e), I guess, and I am not sure what "(e) DAI" means. And for (e), "mean {plus minus} SEM" does not seem applicable. This needs some light revision.

The legend was clarified as followed : "**(d)** Body weight loss, and **(e)** aspect of the feces and presence of occult blood were monitored and used to evaluate Disease activity index in **(f)**."

3 - Figure 1H legend does not say which statistical test was made in the survival experiment in (h) - presumably log-rank? A further comment on the survival statistics: euthanised animals should not be counted towards true mortality when that is what is recorded as an 'event'. They should be right-censored. However, in this case, reaching the euthanasia criterion is just as good an indicator of health as mortality itself. So, simply by changing the Y axis from 'survival' to 'event-free survival' (or something to that effect), where 'events' are either death or reaching the euthanasia criterion, leaves the analysis as it is, and authors do not need to clarify that figure 1H shows "apparent mortality", as it is straightforward "complication-free survival" (just not entirely orthogonal to weight loss).

The Y axis was changed from 'survival' to "percentage of mice not reaching the euthanasia criterion".

4 - Some density measurements are made unnecessarily on arbitrary units (per field of view) - this should be simple to report in absolute measures (i.e. area of tissue screened or, better still, length of epithelium screened).

Because the area of tissue can vary significantly between damages, regenerating and undamaged tissue, we reported the length of epithelium screened as suggested : "per 800um tissue screened" in Fig S1c and Fig 2b.

5 - Figure 2E should read "percent involvement"

This has been corrected.

6 - Figure 2J should read "lipocalin..."

This has been corrected.

7 - In section "CMG2 Is Dispensable for YAP/TAZ-Mediated Reprogramming to Fetal-Like Stem Cells", the authors write ""We measured the mRNA levels of two additional YAP target genes, Cyr61 and CTGF...". I presume the "additional" is because Ly6a is also a target of YAP/TAZ, but if the reader does not know, it is puzzling. I would suggest to make this link explicit.

We added : "In addition to the fetal-like stem cell marker Ly6a, which is a YAP/TAZ target gene, we measured the mRNA levels of two others YAP target genes, Cyr61 and CTGF"

8 - In Figures S2, 3 and S3, I think that the measures expressed as "% of homeostatic X in WT" really mean "% of average homeostatic X in WT". This should be made clear somewhere.

We added: "Dotted line represents the average homeostatic levels of Cmg2 WT" in figure legends

9 - In panel C, the nature of the data is not entirely clear. First, the corresponding part of the legend says "Representative images of n=4 mice per genotype" which I presume should refer to panel B. Then, the graph plots 4 data points, which suggests that they correspond to 4 mice - but how many fields of view? Also, the violin plot outline is not described - I presume it captures all the data points from the coarse-grained pixel analysis, but it should be clarified.

It was modified as suggested : "(c) Results are presented as violin plot of the Ly6a mean intensity of all data points from the coarse-grain analysis. Each symbol represents the mean per mice of n=4 mice per condition. Results are mean \pm SEM. Dotted line represents the average homeostatic levels of Cmg2WT. P values obtained by two-tailed unpaired t test."

10 - In Figure 3H and 3I, I would suggest to add the 7+3 timepoint where the data come from.

We unfortunately do not understand the suggestion of the reviewer, given that these panels show the 7+3 time point.

11 - In section "CMG2 Is Critical for Restoring the Lgr5+ Intestinal Stem Cell Pool", the authors say "...The mRNA levels of ... LRP6, β -catenin (Fig. S3a-b), and Wnt ligands (Wnt5a, 5b, and 2b) were comparable between the colons of Cmg2WT and Cmg2KO mice (Fig. S3c)..." without clarifying in which context - one needs to read the figure legend to realise this is "timepoint 7+3". I suggest to add "in the recovery phase" or "in regenerating colons" or something shorter, just to guide the reader.

We added : "Initially, we quantified the expression of key molecular components involved in Wnt signaling in mice colon 3 days after DSS withdrawal using qPCR."

12 - Like with the previous point, it is not clear when the immunohistofluorescence of B-catenin is made - not even in the legend, as far as I could see. The only hint is that authors say "the nuclei of cells in the atrophic crypts of Cmg2KO..." with 'atrophic' probably indicating again the 7+3 timepoint.

We have changed the text and now mention "*Next, we analyzed β -catenin activation in the colon of Cmg2WT and Cmg2KO mice during the recovery phase.*"

13 - A typo in the discussion: tuning for tuning.

This has been corrected.

14 - In the discussion, the authors talk about the 'CMG2' protein (all caps - formatting convention for human proteins) but before they were referring to 'Cmg2' (formatting convention for mouse proteins). That is fine but some of the statements where "CMG2" is used clearly refer to observations made in the mouse.

We have now used Cmg2, whenever referring to the mouse protein.

15 - Typos in methods: "antigen retrieval by treating [with] Proteinase K"; "Image acquisition and analyze [analysis]"; "All details regarding code<s> used for immunofluorescence analysis".

This has been corrected.

Reviewer #2

We are very pleased to read that the reviewer found the study *“overall well designed, meticulously carried out, and with clear and convincing results that are most reasonably and thoughtfully interpreted”*.

For this reader, one additional thought comes to mind. If I understand the field correctly it would be informative to know with greater confidence where - in what cell type, epithelial or mesenchymal - the CMG2-LRP6-WNT interaction occurs.

This point was also raised by Reviewer 1, and we have now added a new Figure 1, that describes Cmg2 expression in the gut, based both on from publicly available RNA-seq datasets and our RNAscope experiments on Cmg2^{WT} mice. Of note, we unfortunately have never managed to detect Cmg2 protein expression by immunohistochemistry of mouse tissue with any of the antibodies available, commercial or generated in the lab.

After injury the CMG2-KO mouse epithelium exhibits defective WNT signal transduction - as evidenced by failure of β -catenin to translocate into the nucleus. At first glance, this result is a disconnect with the paper by van Rijn that claims the defect in Hyaline Fibromatosis Syndrome cannot be due to loss of CMG2 expression/function in the barrier epithelial cell - a claim based on the mostly normal phenotypes of human CMG2 KO duodenal organoids. But the human organoids studied in the van Rijn paper, like all others, are established and cultured in very high WNT conditions, perhaps obscuring the lack of the CMG2-LRP6-WNT interaction. And in fact, the phenotypes of these human CMG2-KO duodenoids were not entirely normal - the CMG2-KO stem-like organoids (even when cultured in high WNT/R-spondin conditions) developed abnormal intercellular blisters consistent with a defect in epithelial structure/function - of unknown cause and not investigated.

We thank the reviewer for raising this point and we fully agree. We now specify in the text that the human CMG2-KO duodenoids showed blisters, indeed consistent with a defect in epithelial structure/function, and that they were grown on high Wnt media which likely obscure the CMG2 requirement.

I think it would be informative to prepare colon organoids (and duodenoids) from WT and CMG2-KO mice to quantify their WNT dependency during establishment and maintenance of the stem-like (and WNT-dependent) state. If CMG2 acts within the epithelial cell to affect WNT signaling (regardless of WNT source), organoids prepared from colons of CMG2-KO mice would require more WNT in culture media to establish and maintain the stem cell proliferative state - when compared to organoids prepared from WT mice. This can be quantified (and confirmed molecularly by transgene expression if successful). Enhanced dependency of high concentrations of exogenous WNT would be evidence for a primary defect in WNT-(LRP2)-

CMG2 signal transduction localized to the epithelial barrier cell - thus addressing the apparent discrepancy with the van Rijin paper - and for my part, advancing the field. And the discovery of a defect in the epithelium itself for WNT signal transduction would implicate a biologically most plausible mechanism for development of protein losing enteropathy.

By no means do I consider these experiments to be required for publication (especially if considered to be incremental or already defined - WNT-CMG2 is not my field of research). This study already makes a meaningful contribution to the field as I state above. But in the absence of new experimentation, the issue should probably be discussed in greater depth.

We are working out conditions to grow colon organoids that from WT and *Cmg2* KO mice, indeed playing around with the concentrations of Wnt in the various media to identify those that would best mimic the regeneration conditions. This is indeed a study in itself. We have however included a discussion on this point in the manuscript as suggested.

Reviewer #3:

We thank the reviewer for her/his insightful comments.

The premise is that the causative germline mutated gene, CMG2/ANTRX2, may have a functional role in colonic epithelium in addition to controlling the ECM composition. There is little background information but one study has shown no primary defect in epithelial organoids grown from patients with the syndrome. This leads the authors to wonder if non-homeostatic, conditions might reveal a function role for the gene in regeneration.

Reviewer 2 commented on the fact that *“human organoids studied in the van Rijin paper, like all others, are established and cultured in very high WNT conditions, perhaps obscuring the lack of the CMG2-LRP6-WNT interaction. And in fact, the phenotypes of these human CMG2-KO duodenoids were not entirely normal - the CMG2-KO stem-like organoids (even when cultured in high WNT/R-spondin conditions) developed abnormal intercellular blisters consistent with a defect in epithelial structure/function - of unknown cause and not investigated”*.

We have now added a discussion on this point in the manuscript.

The authors' approach to test the hypothesis is to use a mouse germline knockout model and to induce colitis and regeneration by the established protocol of introducing dextran sodium sulfate (DSS) into the drinking water for five days. In brief there is no phenotype apparent in the untreated knockout (KO) but these animals show a more severe response to DSS that requires them to be killed by 10 days after the start of treatment. This effect following phenotypic characterisation of the colonic epithelium is interpreted as showing the CMG2 is a Wnt modifier required for the restoration of the intestinal stem cell population in the final stages of repair.

The experiment and analysis seem reasonably well executed - although a few specific comments follow below. The narrative is simple and easy to understand. However, there are significant caveats that cast doubts on the interpretation made that loss of CMG2 impairs the transition of colonic epithelial cells from a fetal like state to adult ISCs.

1. First there is only a single approach and single type of experiment performed. There is a lack of independent validation of the phenotype and how it is mediated.

We do not fully understand what type of independent validation of the phenotype the reviewer would have liked to see. Is it the induction of intestinal damage using a stress other than DSS?

2. The DSS dose in this kind of experiment is often determined empirically in individual units. Here the 3% used is within published range but at upper end. The control animals show a typical response with symptoms of colitis worsening for 2-3 days after the removal of DSS and then recovery commonly over another 5-7 days. Here the CMG2 KO mice fail to recover and are killed by 9 or 10 days. The authors attempt to exploit the time course by identifying normal initial (7days) and defective late (10days) repair phases in KO animals when compared to controls. It is from this comparison that conclusions are drawn. However, the alternative interpretation might be that the epithelium of KO animals is so badly damaged, and indeed non-existent (from viewing Fig2a), that it is incapable of mounting any other response other than death and that the profiling shown is of an epithelium in extremis. The repair capability and dynamics of the KO would have been better tested under more moderate DSS challenge, if this experiment had been regarded as a pilot rather than as definitive.

The choice of 3% DSS was in fact based on a pilot experiment. As now shown in Fig. S4, we tested different concentrations and found that 3% DSS was the lowest concentration that reliably induced the full spectrum of colitis-associated symptoms, including significant body weight loss, diarrhea, rectal bleeding (summarized in the Disease Activity Index), as well as macroscopic signs such as colon shortening and spleen enlargement. Based on these criteria, we selected 3% DSS for the study described in the manuscript.

In this model, WT mice showed a typical progression: body weight stabilized rapidly after DSS withdrawal, with resolution of diarrhea and rectal bleeding. Histological analysis at day 9 revealed signs of epithelial regeneration, including hypertrophic crypts and increased epithelial proliferation.

In contrast, *Cmg2*^{KO} mice failed to initiate this recovery phase. Clinical signs such as weight loss, diarrhea, and bleeding persisted after DSS withdrawal, ultimately necessitating euthanasia at day 9–10 due to humane endpoint criteria. Unfortunately, this prevented us from exploring later timepoints to determine whether regeneration was delayed or completely abrogated in the absence of *Cmg2*.

Regarding the severity of epithelial damage, as raised by Reviewer 1, we now provide detailed histological scoring in the supplementary data. This analysis shows that the severity of inflammation and crypt damage was similar between WT and KO animals, as were inflammatory markers such as Lipocalin-2. The key difference lies in the extent of tissue involvement. While the lesions in WT mice were more localized, *Cmg2*^{KO} mice displayed widespread and diffuse damage with no sign of regeneration as shown by the absence of hypertrophic crypts and a marked reduction in both epithelial coverage and proliferative cells. Importantly, at day 7, the percentage of epithelial and proliferating cells was comparable between genotypes, further supporting the idea that *Cmg2*^{KO} mice failed to initiate this recovery phase and present a defective repair response.

3. The animals used were young (8 weeks) and lacked any obvious defect in collagen deposition. Does this change with treatment? Even if not, is it possible that there is a defect in peristalsis or transit time of gut contents, resulting in longer dwell times and higher effective dose of DSS to the KO epithelium?

Collagen deposition, particularly of collagen VI, is known to increase in response to intestinal injury and plays a critical role in promoting tissue repair following DSS-induced damage (Molon et al., PMID: 37272555). As suggested, we investigated whether *Cmg2*^{KO} mice exhibit abnormal collagen VI accumulation following DSS treatment.

Our results show that, consistent with published data, WT mice exhibit a marked increase in collagen VI expression during the acute phase of colitis, with levels returning toward baseline following DSS withdrawal. A similar expression pattern was observed in *Cmg2*^{KO} mice, with no significant differences in *Col6a1* mRNA levels between WT and KO animals throughout the entire time course of the experiment. This observation was further confirmed at the protein

level by western blot and immunohistochemistry analyses, suggesting that the impaired regenerative capacity observed in *Cmg2*^{KO} mice is independent of Collagen VI.

Regarding the possibility of altered peristalsis or intestinal transit time contributing to increased DSS exposure in KO mice, this is indeed a possibility. Although we did not directly measure gut motility in this study, we did not observe any signs of intestinal obstruction or fecal retention in *Cmg2*^{KO} mice. Indeed, during the experiment, animals were single caged for 30min in order to collect feces and no difference in the amount of feces collected was observed between WT and KO mice, arguing against a substantial difference in transit time (see figure below). The possible altered peristalsis and these observations are now mentioned in the discussion.

4. Is CMG2 RNA and protein expressed in the colonic epithelium? It is not indicated or tested in the submitted manuscript. This reviewer struggled to find evidence, notably it did not seem to be referenced in the organoid paper they reference in introduction (ref 13).

This very valid point was also raised by Reviewers 1 and 2. The expression pattern of *Cmg2* in the gut has indeed not been characterized and is essential to understanding its function. To address this gap, we added a figure (Fig. 1) providing data from publicly available RNA-seq datasets and from our RNAscope experiments on *Cmg2*^{WT} mice. Of note, we unfortunately have never managed to detect *Cmg2* protein expression by immunohistochemistry of mouse tissue with any of the antibodies available, commercial or generated in the lab.

Specific comments:

Figure 3 c-e and associated text are confusing. In c the Y scale seems inappropriate to show percentages up to 15,000%.

In this graph values are normalized to homeostatic level of WT mice which represent 100%

In d and e the use of percentages may be correct. However, it is claimed in text that *Cty61* and *CTFG* are upregulated in the KO. That is not what the plots appear to show as they compare to WT untreated cells, in which case the KO have not downregulated these genes in the way the controls have.

As clarified in the text, under regenerative conditions, a transient activation of YAP signaling is crucial to induce a fetal-like reversion of intestinal stem cells. However, in a subsequent phase, the downregulation of YAP and the reactivation of Wnt signaling are necessary to complete intestinal regeneration. Several studies have highlighted a strong interplay between the Wnt and YAP pathways, suggesting that their coordinated regulation is essential for effective gut repair. Nevertheless, the precise mechanisms governing this interaction remain incompletely understood.

In our model, this critical transition—YAP downregulation and Wnt reactivation—appears to be impaired. CMG2 may either hinder Wnt reactivation directly, or lead to sustained YAP signaling, which in turn suppresses activation of the Wnt pathway. Further studies, using in-vivo model and organoid models, will be necessary to understand the mechanistic role of *Cmg2* in this regulatory process.

A precision of the figure has been updated as followed: both of which were significantly upregulated in the injured colons of *Cmg2*^{KO} mice compared to DSS-injured *Cmg2*^{WT} mice

****Referees cross-commenting****

Rev2

Points 1 and 2 made by Referee 1 (and point 4 of Referee 3) appear most reasonable, and if not already done should be.

We have indeed addressed these 2 points.

I also noted the more severe morphology of DSS damaged epithelium shown in Fig 2a noted by Referee 3 - and this I agree is a confounding factor. [...] For my part, the concern is understandable but likely not operating in a confounding way. And the evidence for the reprogramming of the damaged epithelium into "fetal-like stem cells" (the 1st step in restitution of lost stem cells) occurs in both WT and KO mice - and these data are strong. For this reader, the block convincingly shows up for KO mouse at the WNT dependent step

The representative image has been updated, and a transverse section has been added to better illustrate that, although both epithelium and crypt structures can be present, the epithelial morphology differs significantly. Indeed, the regenerating epithelium of *Cmg2*^{WT} mice displays a thick epithelial layer with well-polarized epithelial cells, whereas in *cmg2*^{KO} mice, the epithelium appears atrophic, characterized by a thinner epithelial layer and elongated epithelial cells.

Rev 3

This reviewer remains sceptical. I agree the authors performed the experiment well to confirm that DSS dosing was as equivalent as possible across the study. But DSS acts to induce colitis because it is concentrated in the colonic lumen as water is absorbed. Also ECM responses and remodelling are a central part of colitis models. And my concern is that the actual exposure in the KO group is influenced by transit of faeces/DSS is secondary to the known action of CMG2 on collagen deposition. The consequence of this being a protracted damage phase in which a restoration of adult stem cells would not be expected and leading to epithelial failure.

However, we differ. I might propose that the authors are asked to investigate and confirm expression of CMG2 in the epithelium and to repeat the analysis of collagen levels they performed on untreated CMG2 KO mice on colons from CMG2 KO mice having received DSS to see if these differ from controls.

This has now been done.

Rev 1

Both reviewer #2 and reviewer #3 make relevant points, from the point of view of extracting as much biological knowledge as we can from the observations reported in the manuscript.

Reviewer #2 suggestion to use *Cmg2*[KO] organoids to investigate the dependence of Wnt transduction on *Cmg2* is the type of experiments I refrained to propose. However, I think the "skeleton" of the mechanism is there and is reasonably solid. Fleshing it out may well be another paper.

I agree with Reviewer #3 objections to the timing and severity of the DSS damage. However, I am not sure how much they invalidate the main tenet of the paper:

- DSS may affect Cmg2[KO] more severely, but the overall disease score is comparable during the DSS treatment. If this severity was enough to be the main driver of the phenotype, it should have left a mark in the Histological and Disease activity scores. In this regard, I think it would be helpful if the authors provided an expanded version of Figure 2A with examples of the different levels of "Crypt damage" scored, and the proportions for each. This could be in the supplementary material and would balance the impressions induced by a single image.

As suggested, we included a detail of histological score including the crypt damage score in Supplementary Fig 3i showing no significant differences in crypt damage between Cmg2^{WT} and Cmg2^{KO} mice.

- If DSS affected the recovery, this would also be compatible with having a more severe histological phenotype (which is not shown overall, just in Fig 2A) because one would also expect the tissue to attempt regeneration during the 7 days of DSS treatment.

This is an interesting point, and we now allude to this aspect in the manuscript.

- The only objection that I find difficult to argue is the effective duration of the treatment. If indeed peristalsis is affected, it may be that during the 'recovery' phase there is still DSS in the intestine. This could be perhaps verified using a DS detection assay (e.g. <https://arxiv.org/pdf/1703.08663>) on the intestinal contents or the faeces of the mice during the 3-day recovery period.

We have attempted to obtain and purchase Heparin Red to perform this assay. Unfortunately, we have not obtained the reagent, which has never been delivered. We now also mention the following in the Discussion:

One could envision that Cmg2^{KO} mice have a defect in peristalsis resulting in longer dwell times and possibly higher effective dose of DSS to the KO epithelium. We however did not observe any signs of intestinal obstruction or fecal retention in Cmg2^{KO} mice. Animals were single-caged for 30 min to collect feces. We did not observe any difference in amounts collected from WT and KO mice, arguing against a substantial difference in transit time of gut contents. Moreover, if DSS affected the recovery, one would have expected a more severe histological phenotype in the colon of Cmg2^{KO} since the tissue likely already attempts regeneration during the 7 days of DSS treatment. But this was not the case. Therefore, while we cannot formally rule out the presence of residual DSS in Cmg2^{KO} mice during the DSS withdrawal phase, there is currently no indication that this was the case.

I think of what the aim of scholarly publication is, with this paper, and I find myself going back to a statement of the authors' discussion - that this work suggests that infants risking death may be offered (compassionate, I guess) IBD treatment. What does this hinge upon? I think, on the basic observation that diarrhoea (in the mouse model) is not intrinsic but caused by an inflammation-promoting insult. Is this substantiated? I think it is. Could we learn more biology from this disease model, about Wnt and about how ECM affects tissue regeneration? Certainly. Can this learning wait? I believe it can.

We thank the reviewer for this statement.

23rd Jun 2025

Dear Prof. van der Goot,

Thank you for submitting your revised study to EMBO Molecular Medicine following peer-review at Review Commons. We have asked the original reviewers to evaluate your revised manuscript. Unfortunately, reviewer #1 was unavailable; however, reviewer #2 (now reviewer #1) and #3 (now reviewer #2) evaluated your revised manuscript, and also checked your answers to the third reviewer.

As you will see from the reports below, they are satisfied with the revisions and I will therefore be able to accept your manuscript once the following editorial concerns are addressed:

1/ Manuscript text:

- Please remove the grey highlighted text, and only indicate in track changes mode any new modification in the text.
- Please provide up to 5 keywords.
- The manuscript sections should be in the following order: Title page - Abstract & Keywords - Introduction - Results - Discussion - Methods - Data Availability - Acknowledgments - Disclosure Statement & Competing Interests - References - Figure Legends - (Main Tables with legends if applicable) - Expanded View Figure Legends.
- Summary should be renamed 'Abstract'.
- Supplementary information (p33-37) needs to be removed, and merged with the methods and/or the appendix.
- Materials and Methods should be renamed Methods:
 - o Please download and fill our Reagents and Tools Table template (.docx), which you can find in our author guidelines: <https://www.embopress.org/page/journal/14693178/authorguide#structuredmethods>. When submitting your revised manuscript, please do not include the Reagents and Tools Table in the Methods section of the manuscript but upload it as a separate file choosing the file type "Reagent Table".
 - o Cells: please indicate whether the cells were authenticated and tested for mycoplasma contamination.
 - o Statistics: please provide a statement on sample size, blinding and randomization, and inclusion/exclusion criteria.
- Data and Code Availability: please rename "Data Availability". Data should be publicly available before acceptance.
- Declaration of interests: please rename "Disclosure statement and competing interests".
- Author contributions: CRediT has replaced the traditional author contributions section because it offers a systematic machine readable author contributions format that allows for more effective research assessment. Please remove the Authors Contributions from the manuscript and use the free text boxes beneath each contributing author's name in our system to add specific details on the author's contribution.
- Acknowledgements: the funding information provided in the manuscript should match the information provided in the submission system (currently, grant number 310030_214874 for Swiss National Science Foundation is missing in the manuscript, and the Gelù Foundation is missing in the submission system).
- Please reformat the references in alphabetical order; DOIs should only be used for preprints and datasets that have not been published yet.
- Our journal encourages inclusion of *data citations in the reference list* to directly cite datasets that were re-used and obtained from public databases. Data citations in the article text are distinct from normal bibliographical citations and should directly link to the database records from which the data can be accessed. In the main text, data citations are formatted as follows: "Data ref: Smith et al, 2001" or "Data ref: NCBI Sequence Read Archive PRJNA342805, 2017". In the Reference list, data citations must be labeled with "[DATASET]". A data reference must provide the database name, accession number/identifiers and a resolvable link to the landing page from which the data can be accessed at the end of the reference. Further instructions are available at .

2/ Figures:

- Please provide individual production quality figure files for the main and EV figures.
- We replaced Supplementary Information with Expanded View (EV) Figures and Tables that are collapsible/expandable online. EV Figures should be cited as 'Figure EV1, Figure EV2" etc... in the text and their respective legends should be included in the main text after the legends of regular figures. For the figures that you do NOT wish to display as Expanded View figures, they should be bundled together with their legends in a single PDF file called *Appendix*, which should start with a short Table of Content. Appendix figures should be referred to in the main text as: "Appendix Figure S1, Appendix Figure S2" etc.
- Please note: panel e is labeled twice in Figure 1 (label for panel f is missing).
- During our standard figure check, we noted potential image duplication between Figures S2E and S5I. Please carefully check, correct if needed and provide an explanation. Additionally, kindly note that figure re-use should be indicated in the figure legend (i.e. Appendix Figure 7B and Appendix Figure 7D).
- Please address the queries from our data editors:
 1. Please note that the exact p values are not provided in the legends of figures 2C, D, F, G, H; 3G, H; 4H
 2. Please note that the white dotted borders are not defined in the legend of figure 4I. This needs to be rectified.

3/ At EMBO Press we ask authors to provide source data for the main figures. Our source data coordinator will contact you to discuss which figure panels we would need source data for and will also provide you with helpful tips on how to upload and

organize the files.

4/ Please provide a complete author checklist, which you can download from our author guidelines (<https://www.embopress.org/page/journal/17574684/authorguide#submissionofrevisions>). Please insert information in the checklist that is also reflected in the manuscript. The completed author checklist will also be part of the RPF.

5/ Please provide 'The paper explained', which should have the following structure:

6/ Please provide a synopsis text: It should include a short stand first (maximum of 300 characters, including space) as well as 2-5 one-sentences bullet points that summarizes the paper (maximum of 30 words / bullet point). Please upload the synopsis as a separate document.

Please also upload the visual abstract as a separate file in jpeg, TIFF or png format (550 pixels wide x 200-600 pixels high). A cropped portion of this image will serve as thumbnail for the table of content on our webpage.

7/ As part of the EMBO Publications transparent editorial process initiative (see our Editorial at <http://embomolmed.embopress.org/content/2/9/329>), EMBO Molecular Medicine will publish online a Review Process File (RPF) to accompany accepted manuscripts.

This file will be published in conjunction with your paper and will include the anonymous referee reports, your point-by-point response and all pertinent correspondence relating to the manuscript. Let us know whether you agree with the publication of the RPF and as here, if you want to remove or not any figures from it prior to publication.

I look forward to receiving your revised manuscript.

Yours sincerely,

Lise Roth

***** Reviewer's comments *****

Referee #1 (Remarks for Author):

the paper addresses an important topic with thoughtful and carefully designed and executed studies. Results make an important contribution to the field

Referee #2 (Remarks for Author):

The authors have responded in depth and appropriately to the previous criticisms. The clarifications and additional analyses have resulted in a much improved manuscript. No further concerns.

Rev_Com_number: RC-2025-02862

New_manu_number: EMM-2025-22001-T

Corr_author: van der Goot

Title: Injury-Induced Intestinal Stem Cell Renewal Requires Capillary Morphogenesis Gene 2

The authors addressed the remaining formatting issues.

22nd Jul 2025

Dear Prof. van der Goot,

Thank you for sending your revised files. We are pleased to inform you that your manuscript is accepted for publication and is now being sent to our publisher to be included in the next available issue of EMBO Molecular Medicine.

Yours sincerely,

Lise Roth
